# Endocannabinoid dynamics gate spike-timing dependent depression and potentiation

Yihui Cui[1,2], Ilya Prokin[3,4], Hao Xu[1,2], Bruno Delord[2,5], Stephane Genet[2,5], Laurent Venance[1,2*†], Hugues Berry[3,4*†]

[1]Center for Interdisciplinary Research in Biology, College de France, INSERM U1050, CNRS UMR7241, Labex Memolife, Paris, France; [2]University Pierre et Marie Curie, ED 158, Paris, France; [3]INRIA, Villeurbanne, France; [4]LIRIS UMR5205, University of Lyon, Villeurbanne, France; [5]Institute of Intelligent Systems and Robotics, Paris, France

**Abstract** Synaptic plasticity is a cardinal cellular mechanism for learning and memory. The endocannabinoid (eCB) system has emerged as a pivotal pathway for synaptic plasticity because of its widely characterized ability to depress synaptic transmission on short- and long-term scales. Recent reports indicate that eCBs also mediate potentiation of the synapse. However, it is not known how eCB signaling may support bidirectionality. Here, we combined electrophysiology experiments with mathematical modeling to question the mechanisms of eCB bidirectionality in spike-timing dependent plasticity (STDP) at corticostriatal synapses. We demonstrate that STDP outcome is controlled by eCB levels and dynamics: prolonged and moderate levels of eCB lead to eCB-mediated long-term depression (eCB-tLTD) while short and large eCB transients produce eCB-mediated long-term potentiation (eCB-tLTP). Moreover, we show that eCB-tLTD requires active calcineurin whereas eCB-tLTP necessitates the activity of presynaptic PKA. Therefore, just like glutamate or GABA, eCB form a bidirectional system to encode learning and memory.

*For correspondence: laurent.
venance@college-de-france.fr (LV);
hugues.berry@inria.fr (HB)

†These authors contributed
equally to this work

Competing interests: The
authors declare that no
competing interests exist.

Reviewing editor: Upinder S
Bhalla, National Centre for
Biological Sciences, India

## Introduction

Bidirectional long-term plasticity of synaptic strength (LTD and LTP) underlies multiple forms of learning and memory (*Citri and Malenka, 2008*; *Nabavi et al, 2014*). Bidirectionality is of paramount functional importance since it allows LTP and LTD to reverse each other with time at a single synapse, thus enabling adaptive changes of the synaptic weight. Endocannabinoids (eCBs) have emerged as a major actor in learning and memory because of their powerful influence on synaptic plasticity (*Chevaleyre et al., 2006*; *Heifets and Castillo, 2009*; *Kano et al., 2009*; *Katona and Freund, 2012*). The eCB system is mainly composed of active biolipids (notably 2-arachidonylglycerol, 2-AG and anandamide, AEA) synthesized and released on-demand acting as retrograde neurotransmitters on presynaptic type-1 cannabinoid receptor (CB1R) and postsynaptic transient receptor potential vanilloid-type-1 (TRPV1) (*Piomelli, 2003*; *Piomelli et al., 2007*; *Di Marzo, 2008*; *Alger and Kim, 2011*).

The major neurotransmitter systems, glutamate and GABA, allow bidirectional synaptic plasticity (*Citri and Malenka, 2008*), i.e. the same signaling pathway in the same cell gates the neuron towards potentiation or depression depending on the activity pattern. In contrast, eCBs have been widely described as a powerful unidirectional system that depresses neuronal communication on a short or long timescale. However, recent reports challenge this view and indicate that eCBs could also act as a bidirectional system for synaptic plasticity. We recently reported the existence of an

**eLife digest** Learning and memory depend on processes that alter the connections – or synapses – between neurons in the brain. For example, molecules called endocannabinoids can alter synapses to decrease the influence that one neuron has on another neuron's activity. This "synaptic depression" is an important mechanism through which the brain can adapt to an experience.

However, recent research also suggests that endocannabinoids might also increase the influence one neuron has on another neuron's activity by strengthening the synaptic connection between neurons. This opposite process is known as synaptic potentiation, and is also important for learning from experience. But how do endocannabinoids manage to produce opposing effects?

Using a combination of electrophysiological recording experiments and mathematical modeling, Cui et al. have now deciphered the molecular mechanisms that govern the action of endocannabinoids at key synapses in rat brain slices. This revealed that both the levels and timing of endocannabinoid release control changes in the strength of the synaptic connections. Electrical stimulations that produced moderate amounts of endocannabinoids over a prolonged period led to synaptic depression. However, stimulation that produced short but large endocannabinoid peaks caused synaptic potentiation. The enzymes that control endocannabinoid levels thus play a crucial role in determining whether a given stimulation leads to the strengthening or weakening of a synaptic connection.

In the type of synapses studied by Cui et al., changes to synaptic strength also depend on another chemical called dopamine. Abnormal dopamine production is implicated in a number of disorders, including Parkinson's disease and addiction. Future work will therefore investigate how dopamine controls endocannabinoid-dependent changes to the strength of synapses.

eCB-mediated spike-timing dependent LTP in the dorsal striatum induced by a low number of paired stimulations and dependent on the activation of CB1R and TRPV1 (*Cui et al, 2015*). We found that few coincident pre- and post-synaptic spikes (5–15) were sufficient to increase synaptic efficacy through a signaling pathway that relies on the activation of CB1R and TRPV1 and on 2-AG elevations. The latter are triggered by coupled postsynaptic rises of calcium and DAG lipase $\alpha$ (DAGL$\alpha$) activity mediated by type-5 metabotropic glutamate receptors (mGluR5), muscarinic M1 receptors and voltage-sensitive calcium channels (VSCCs) (*Cui et al, 2015*). In addition, it has been reported an indirect role of eCBs in promoting LTP at mixed (chemical and electrical) synapses of the goldfish Mauthner cell via intermediary dopaminergic neurons (*Cachope et al., 2007*) or at hippocampal CA1 synapses via a GABA$_A$ receptor-mediated mechanism (*Lin et al., 2011*; *Xu et al., 2012*). Likewise, facilitation of LTP in the hippocampus via eCB-induced presynaptic depression of GABAergic transmission (*Carlson et al., 2002*; *Chevaleyre and Castillo, 2004*; *Zhu and Lovinger, 2007*), and mediation of heterosynaptic short-term potentiation via intermediary astrocytes (*Navarrete and Araque, 2010*) have been reported. There exists a growing body of evidence that paves the way for a bidirectional action of eCBs in synaptic plasticity depending on the activity pattern on either side of the synapse.

In the case of glutamate, the principal mechanism put forward to account for bidirectionality is the calcium-control hypothesis, which states that postsynaptic calcium levels and/or time courses decide the outcome of plasticity (LTP or LTD) (*Shouval et al., 2002*; *Graupner and Brunel, 2012*). However, how eCBs induce both LTD and LTP remains to be elucidated.

Here, combining experimental and computer modeling approaches, we show that the bidirectionality of eCB-dependent STDP in striatum is controlled by eCB-levels: moderate level and prolonged release of eCB lead to LTD while brief releases of high eCB concentration yield LTP. In this aspect, MAG-lipase appears as a key controller of synaptic plasticity. Our results considerably enlarge the spectrum of action of eCBs since they show that eCBs not only promote depression but also potentiation, i.e. they act as a bidirectional system, depending on the regime of activity pattern on either side of the synapse.

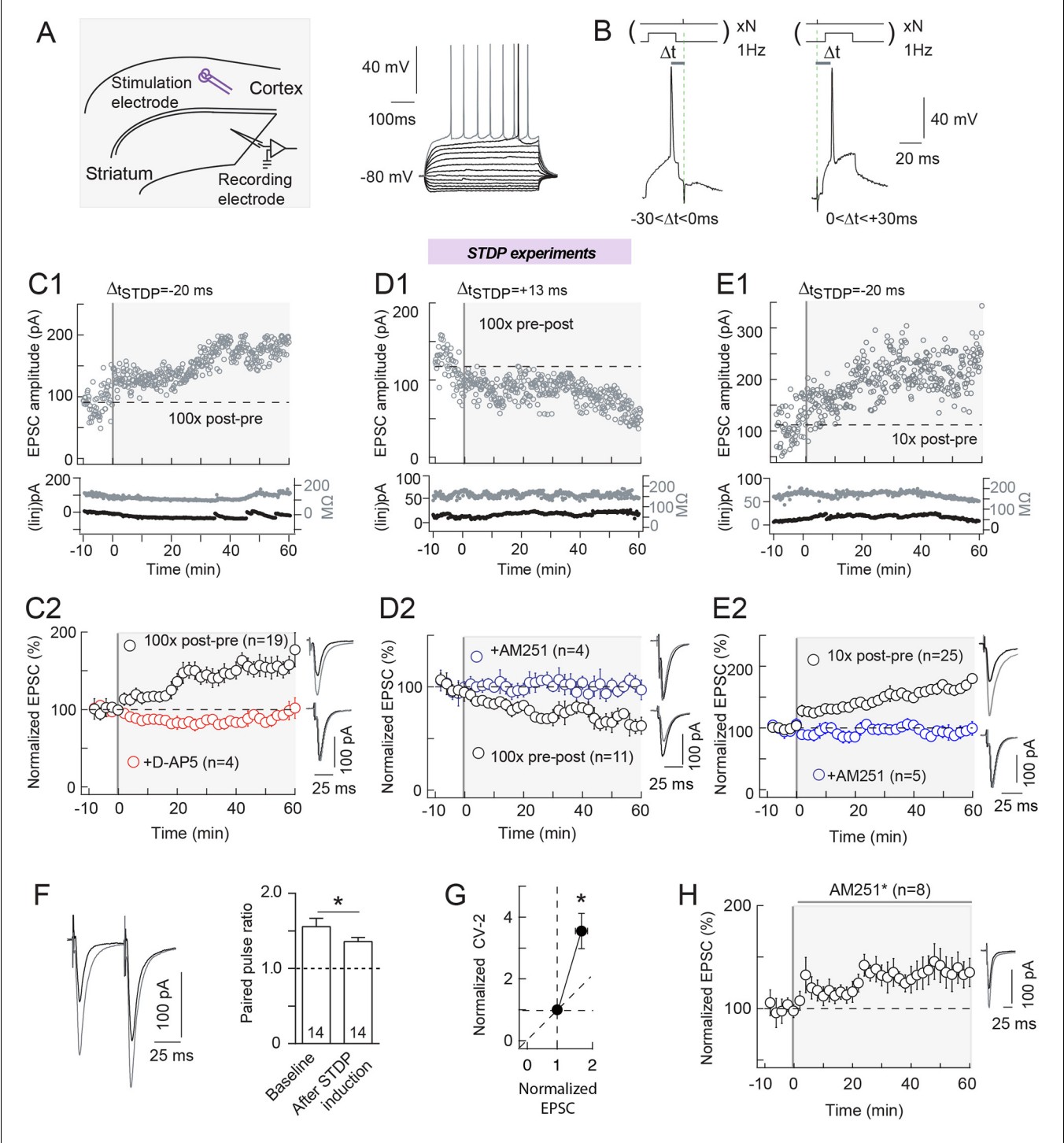

**Figure 1.** Bidirectional endocannabinoid-mediated STDP depends on the number of pairings. (A) Whole-cell recording from the dorsal striatum with the stimulation electrode placed in layer 5 of the somatosensory cortex in horizontal rat brain slice. (B) Experimental design. Extracellular stimulation evoked EPSCs monitored at RMP. During pairings, recordings were switched to current-clamp to allow postsynaptic MSN to fire single action potentials paired with single cortical extracellular stimulations. MSN and cortical stimulation were repeated N times (10 or 100) at 1 Hz. $\Delta t_{STDP}$ indicates the time delay between pre- and post-synaptic stimulations. $-30 < \Delta t_{STDP} < 0$ ms and $0 < \Delta t_{STDP} < +30$ ms refers to post-pre and pre-post pairings, respectively. (C) 100 post-pre pairings induced NMDAR-mediated tLTP. (C1) Example of tLTP induced by 100 post-pre pairings. Top, EPSC strength before and after pairings (before pairings: 91±3 pA; 45–55 min after pairings: 169±2 pA; increase of 87%). Bottom, time courses of Ri (before, 132±1 MΩ; after, 134±2 MΩ; change of 2%) and injected current (Iinj) (before, -2±1 pA; after, -12±2 pA; change of 6% of baseline EPSC amplitude) for this cell. (C2) Summary of tLTP induced by 100 post-pre pairings. 15/19 cells showed significant tLTP. Inhibition of NMDAR with D-AP5 (50 μM, n=4) prevented the

*Figure 1 continued on next page*

*Figure 1 continued*

induction of tLTP; 4/4 cells showed no significant plasticity. The normality of D-AP5 data was assumed (test not passed). (**D**) 100 pre-post pairings induced CB1R-mediated tLTD. (**D1**) Example of tLTD induced by 100 pre-post pairings. Top, EPSC strength before and after pairings (before pairings: 134±2 pA; 45–55 min after pairings: 82±2 pA; decrease of 39%). Bottom, time courses of Ri (before, 156±2 MΩ; after, 157±1 MΩ; change of 1%) and injected current (Iinj) (before, 14±1 pA; after, 20±1 pA; change of 5%) for this cell. (**D2**) Summary of tLTD induced by 100 post-pre pairings. 7/11 cells showed significant tLTD. Inhibition of CB1R with AM251 (3 μM, n=4) prevented the induction of tLTD; 4/4 cells showed no significant plasticity. The normality of AM251data was assumed (test not passed). (**E**) 10 post-pre pairings induced CB1R-mediated tLTP. (**E1**) Example of tLTP induced by 10 post-pre pairings. Top, EPSC strength before and after pairings (before pairings: 112±4 pA; 45–55 min after pairings: 213±4 pA; increase of 90%). Bottom, time courses of Ri (before, 171±2 MΩ; after, 167±1 MΩ; change of 2%) and injected current (Iinj) (before, 10±1 pA; after, 12±1 pA; change of 2%) for this cell. (**E2**) Summary of tLTP induced by 10 post-pre pairings. 21/25 cells showed significant tLTP. Inhibition of CB1R with AM251 (3 μM, n=5) prevented the induction of tLTP; 5/5 cells showed no significant plasticity. Normality was assumed for the ctrl 10x post-pre data (test not passed). (**F-H**) eCB-LTP is maintained by a mechanism located downstream of CB1R activation in the presynaptic terminals. (**F**) Representative EPSCs and summary bar graphs (n=14) of paired-pulse cortical stimulations (50 ms interstimulus interval) illustrate a decrease of facilitation after 10 post-pre pairings. This indicates a presynaptic locus of the eCB-tLTP. (**G**) Mean variance analysis ($CV^{-2}$, n=17) indicates a presynaptic locus of the eCB-tLTP maintenance. (**H**) Summary of tLTP induced by 10 post-pre pairings with application of CB1R inhibitor just after the pairings (AM251*) (7/8 cells showed significant tLTP). This treatment did not prevent tLTP, indicating that the maintenance of eCB-tLTP does rely on the signaling downstream of CB1R. Normality was assumed for the data og $CV^{-2}$ after STDP protocol (test not passed). Representative traces are the average of 15 EPSCs during baseline (black traces) and 50 min after STDP protocol (grey traces). Error bars represent s.d. *p<0.05. ns: not significant.

The following figure supplement is available for figure 1:

**Figure supplement 1.** NMDAR-tLTP relies on CaMKII activity.

## Results

### Endocannabinoids mediate spike-timing dependent LTD and LTP (eCB-tLTD and eCB-tLTP) depending on the number of pairings.

STDP is a major synaptic Hebbian learning rule (*Sjöström et al., 2008*; *Feldman, 2012*) in which synaptic weight changes depend on the time delay $\Delta t_{STDP}$ between presynaptic and postsynaptic paired stimulations: $\Delta t_{STDP}<0$ when post-synaptic stimulation occurs before the paired pre-synaptic one (*post-pre* pairings), whereas $\Delta t_{STDP}>0$ when pre-synaptic stimulation occurs before the post-synaptic one (*pre-post* pairings). Corticostriatal synapses are known to exhibit a bidirectional eCB-dependent STDP in which tLTP or tLTD can be obtained depending on the spike timing ($\Delta t_{STDP}$) but also on the number of pairings ($N_{pairings}$) (*Fino et al., 2005*; *Shen et al., 2008*; *Pawlak and Kerr, 2008*; *Fino et al., 2010*; *Paillé et al., 2013*; *Cui et al., 2015*). In agreement with those reports, we obtained a bidirectional plasticity when we induced STDP with 100 pairings in medium-sized spiny neurons (MSNs): post-pre pairings (-30<$\Delta t_{STDP}$<0 ms) induced tLTP (mean value of the EPSC amplitude recorded 50 min after STDP protocol: 156±15%, p=0.0015, n=19), while pre-post pairings (0<$\Delta t_{STDP}$<+30 ms) induced tLTD (76±8%, p=0.0051, n=11) (*Figure 1A,B, C1-2 and D1-2*). Note that this STDP displays an anti-hebbian polarity in accordance with previous reports (*Fino et al., 2005*; *Fino et al., 2010*; *Schulz et al., 2010*; *Paillé et al., 2013*; *Cui et al., 2015*) but not with other studies (*Pawlak and Kerr, 2008*; *Shen et al., 2008*) at corticostriatal synapses (*Fino and Venance, 2010*). We have previously shown that GABA acts as an Hebbian/anti-Hebbian switch (*Paillé et al., 2013*), so polarity of the corticostriatal STDP depends on whether GABA_A receptor antagonists are applied (Hebbian STDP; *Pawlak and Kerr, 2008*; *Shen et al., 2008*) or not (anti-Hebbian STDP; *Fino et al., 2005*; *Fino et al., 2010*; *Fino and Venance, 2010*; *Cui et al., 2015*; this study). Examples of tLTP and tLTD induced by 100 post-pre and 100 pre-post pairings are shown in C1 and D1, respectively, and the experiment summary in C2 and D2. tLTP was NMDAR-mediated since blocked by the selective NMDAR blocker D-AP5 (50 μM) (99±3%, p=0.7998, n=4) (*Figure 1C2*). while tLTD relied on eCBs because pharmacological inhibition of CB1R with AM251 (3 μM) impaired this plasticity (102±7%, p=0.8108, n=4) (*Figure 1D2*). As recently reported (*Cui et al., 2015*), lowering the number of pairings down to 10 yields tLTP for post-pre pairings (163±12%, p<0.0001, n=25) (*Figure 1E* with an example of LTP induced by 10 post-pre pairings in E1 and the experiment summary in E2) and a lack of significant plasticity for pre-post pairings (97±11%, p=0.3844, n=8). tLTP induced with 10 post-pre STDP pairings was CB1R-mediated since treatment with AM251 (3 μM)

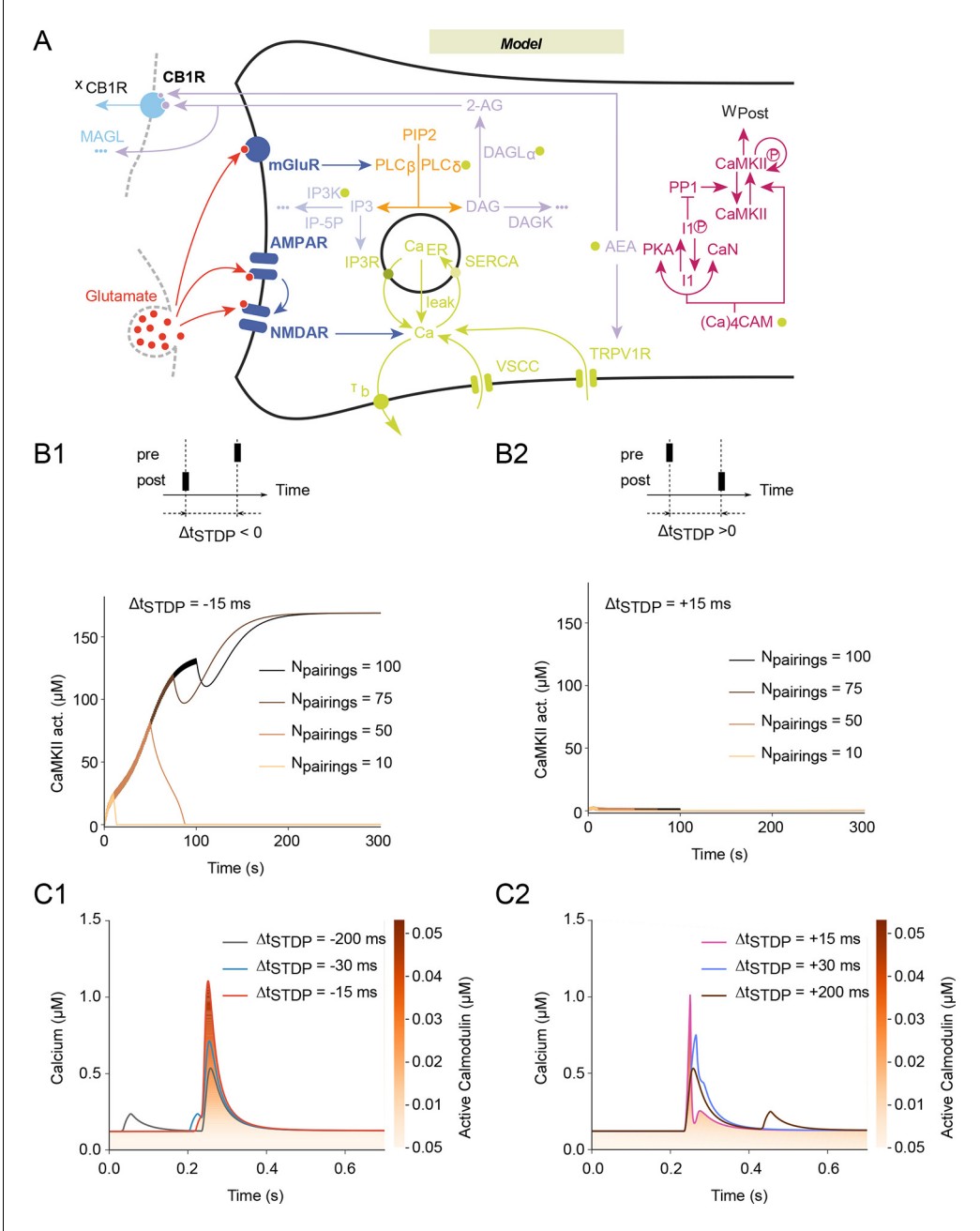

**Figure 2.** Mathematical model predicts NMDAR-tLTP with large numbers of post-pre pairings. (**A**) Scheme of the modeled signaling network. The synaptic weight $W_{total}$ is the product of pre- and postsynaptic weights $W_{pre}$ and $W_{post}$. The NMDAR-based pathway sets $W_{post}$ as the phosphorylation state of postsynaptic CaMKIIα. In the second pathway, coincident activation of phospholipase Cβ by postsynaptic mGluR and calcium entry via VSCC and TRPV1 induces the production of 2-AG and AEA. 2-AG, and to a lower extent AEA, activates CB1R ($x_{CB1R}$ is the fraction of non-desensitized CB1R), which then modulates the presynaptic weight, $W_{pre}$. Color code: glutamate receptors: dark blue; PLC pathway: yellow; IP3 pathway: powderblue; calcium pathways: green (green disks indicate calcium-dependent steps); DAGLα pathway: lavander; AEA pathway: light blue; CB1R pathway: blue. Abbreviations: PIP2: phospatidylinositol 4,5-biphosphate; DAG: diacylglycerol; IP3: inositol-1,4,5-triphosphate; PLCβ/δ: phospholipase C β/δ; DAGK: diacylglycerol kinase; IP-5P: inositol polyphosphate 5-phosphatase; IP3K: IP3-kinase; DAGLα: diacylglycerol lipase α; B/BCa: free/bound endogeneous calcium buffer; IP3R: IP3-receptor channel; SERCA: sarcoplasmic/endoplasmic reticulum calcium ATPase; Ca$_{ER}$: calcium in the endoplasmic reticulum; (Ca)$_4$CaM: fully bound calmodulin; CaN: calcineurin aka PP2B; PKA: protein kinase A; I1p/I1: phosphorylated/unphosphorylated protein phosphatase 1 inhibitor 1 (DARPP-32 in striatal output neurons); PP1:

*Figure 2 continued on next page*

*Figure 2 continued*

protein phosphatase 1; pCaMKII/CaMKII: phosphorylated/unphosphorylated CaMKII; DAGK: diacylglycerol kinase; MAGL: monoacylglycerol lipase; the '. . .' sign indicates transformation into products that are considered not to interfere with the other interactions of the model. (B) Corresponding changes in the levels of active CaMKII starting from the down (non-activated) state. The number of pairings, $N_{pairings}$, is indicated for 1 Hz pairings at spike-timing $\Delta t_{STDP}$=-15 (B1) or +15 (B2) ms. (C) Intracellular calcium changes for the first pairing in post-pre (C1) or pre-post (C2) pairing protocols. The colorcode shows the corresponding amount of calmodulin activation according to the colorbar.

resulted in an absence of significant plasticity (88±11%, p=0.3073, n=5) (*Figure 1E2*). Based on this eCB-dependence, we refer to the tLTP triggered by 10 post-pre pairings as eCB-tLTP.

Location of CB1R at the presynaptic terminals of the corticostriatal pathway (*Katona and Freund, 2012*) suggests that the locus of eCB-tLTP maintenance would likely be presynaptic. First, we applied presynaptic paired pulses with 50 ms interpulse interval, known to induce a significant EPSC paired-pulse facilitation (PPF) in MSNs, (*Goubard et al., 2011*) before and after STDP pairings. We observed a significant decrease of the PPF after the STDP pairings (PPF$_{plasticity/baseline}$=0.872±0.044, p=0.0470, n=14) (*Figure 1F*), which indicates a presynaptic locus of eCB-tLTP. Second, using the mean variance analysis of EPSCs, we found a $CV^{-2}$ value of 3.6 ± 0.6 (p=0.0008, n=17), which confirmed a presynaptic maintenance of eCB-tLTP (*Figure 1G*). To further distinguish between induction and maintenance loci, we performed experiments in which we applied the CB1R antagonist AM251 just after the STDP pairings, and we still observed significant tLTP (146±12%, p=0.0092, n=8) (*Figure 1H*) whereas AM251 applied during the protocol prevented tLTP (*Figure 1E2*). This indicates that eCB-tLTP is maintained by a mechanism located downstream of CB1R activation in the presynaptic terminals.

## A mechanism accounting for eCB-LTP induction for low numbers of pairings

We then questioned how eCBs could mediate either potentiation or depression, depending on the activity pattern of either side of the synapse. To address this question, we built a realistic mathematical model of the molecular mechanisms of corticostriatal synaptic plasticity (*Figure 2A*). Our model is based on the two signaling pathways involved in corticostriatal STDP induced by 100 pairings: NMDAR- and CB1R-signaling (*Pawlak and Kerr, 2008*; *Shen et al., 2008*; *Fino et al., 2010*; *Paillé et al., 2013*). NMDAR-tLTP is CaMKII-dependent since we found that pharmacological inhibition of CaMKII with KN62 (3 µM) blocked NMDAR-tLTP (88 ± 11%, p=0.3324, n=6) (*Figure 1—figure supplement 1*). We thus combined in the model a first signaling pathway leading from NMDAR to calmodulin and CaMKII with a second, distinct one that assembles mGluR and cytosolic calcium to eCB production and the resulting activation of CB1R (*Figure 2A*). Most of the parameter values were restricted by previous experimental measurements (*Supplementary file 1*).

In the model, the total synaptic weight ($W_{total}$) is given by the product of presynaptic ($W_{pre}$) and postsynaptic ($W_{post}$) contributions (see Methods). The postsynaptic contribution to the synaptic weight, $W_{post}$ is taken proportional to the amount of CaMKII activated by the NMDAR pathway. This part of our model (from *Graupner and Brunel, 2007*) exhibits bistable dynamics between a down state where CaMKII is inactive and an up state where CaMKII is highly activated (*Figure 2B1*). Transitions between those two states therefore emulate transitions between no plasticity (down state) and NMDAR-tLTP (up state). The time scale of CaMKII dephosphorylation after a pairing being larger than the period between two successive pairings (1 sec), the amounts of activated CaMKII progressively accumulates with the number of pairings. Importantly, the level of activated CaMKII needs 50–60 post-pre pairings (with $\Delta t_{STDP}$=-15 ms) to reach the threshold between the up and down states (*Figure 2B1*). As a result, $W_{post}$ converges to the up state (potentiation) only when $N_{pairings}$>50 post-pre pairings, thus emulating the experimental observations of NMDAR-dependent LTP and its dependence on the number of pairings (*Cui et al., 2015*). For pre-post pairings, the calcium response after each pairing activates less of the CaMKII-activating calmodulin (*Figure 2C*) so the amount of activated CaMKII never reaches the threshold for the up state (*Figure 2B2*). Thus, the

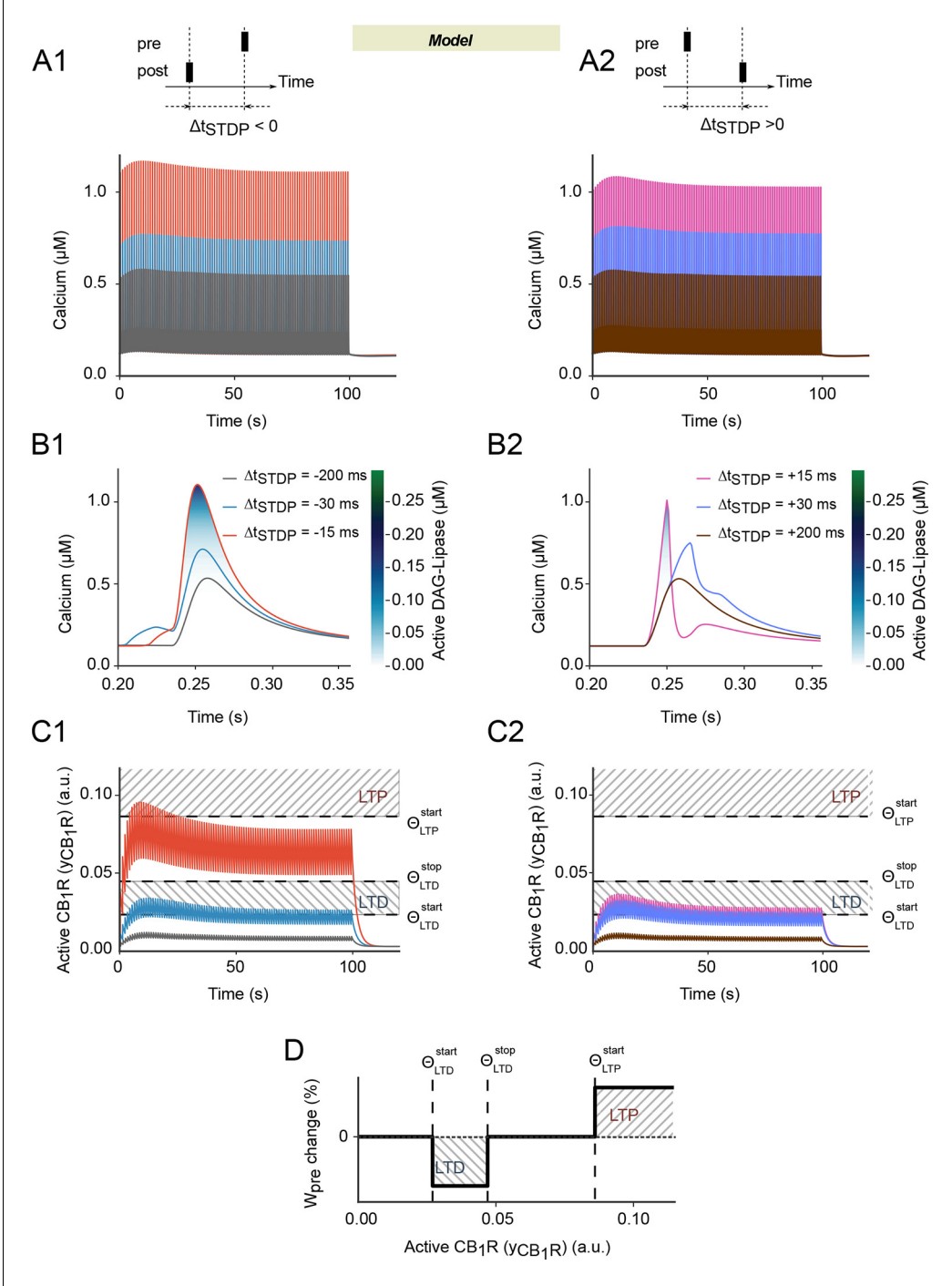

**Figure 3.** Spike-timing dependence in the endocannabinoid-signaling part of the model. (A-C) Predicted dynamics of cytoplasmic calcium and CB1R activation for post-pre (first column) or pre-post pairings (second column): in our model, the postsynaptic calcium peaks (A, B) are slightly more width at large calcium values with post-pre (A1-B1) than pre-post pairings (A2-B2). As a consequence, DAGLα activation (color-coded in B) and the resulting CB1R activation, $y_{CB1R}$ (C) is larger too. The biphasic envelope of the calcium peak amplitude with the number of pairings (A) is amplified as a marked biphasic envelope for the amplitude of $y_{CB1R}$. (C) As a result, whatever the stimulation, the amplitude of the $y_{CB1R}$ peaks first increases for the first 10–20 pairings, then decreases to converge to roughly constant amplitude. But $y_{CB1R}$ reaches large values only for short post-pre pairings (C1). This particular dynamics of $y_{CB1R}$ during the stimulations suggests a possible explanation to the bidirectional characteristics of eCB-dependent plasticity, where the presynaptic contribution to the synaptic

*Figure 3 continued on next page*

*Figure 3 continued*

weight $W_{pre}$ depends on the magnitude of $y_{CB1R}$. (D) $W_{pre}$ decreases (LTD) when $y_{CB1R}$ reaches an intermediate range whereas it increases (LTP) if $y_{CB1R}$ overcomes a LTP threshold. The corresponding thresholds and ranges are reported in C1-2 as dashed lines and hashed boxes, respectively.

model predicts no NMDAR-dependent for pre-post pairings ($0 < N_{pairings} < 100$ at 1 Hz, *Figure 2C2*), in agreement with experimental observations (*Cui et al., 2015*).

Within a wide parameter range, the amplitude of the calcium peaks triggered by each paired stimulation shows a peculiar biphasic envelope (*Figure 3A*): calcium first increases for the first 10–20 pairings then decreases afterwards, until it reaches constant amplitude after 50 pairings. During the first 10–20 pairings, repeated activation of mGluRs progressively increases the quantity of IP$_3$, which contributes an extra influx of calcium from the endoplasmic reticulum. This boost of cytoplasmic calcium however progressively disappears when N$_{pairings}$ increases further because the concentration of calcium in the endoplasmic reticulum decreases. Moreover, after each pairing, the width of the postsynaptic calcium peak in the model is larger with post-pre than pre-post pairings (*Figure 3B*). As a consequence, the fraction of calcium-activated DAGLα is significant only for small values of |Δt$_{STDP}$| (<25 ms) and larger for post-pre than pre-post pairings. As a result, the biphasic envelope of the calcium peak amplitude with N$_{pairings}$ (first increase, then decrease) is transmitted to the amplitude of eCB transients and, ultimately, to CB1R activation ($y_{CB1R}$). The biphasic envelope is even more marked at the level of CB1R activation because of CB1R desensitization that amplifies the decay above 20 pairings. *Figure 3C* illustrates the dynamics of CB1R activation in the model. In all cases, the amplitude of the CB1R activation peaks first increases for the first 10–20 pairings, then decreases to converge to constant amplitude. $y_{CB1R}$ reaches large values only for short post-pre pairings (Δt$_{STDP}$ around -15 ms) while even short pre-post pairings (0<Δt$_{STDP}$<10 ms) do not give rise to such large amplitude peaks.

This peculiar dynamics of $y_{CB1R}$ brings a plausible explanation to the bidirectional features of eCB-dependent plasticity. Under this scenario, $W_{pre}$ depends on the magnitude of $y_{CB1R}$ so that whenever $y_{CB1R}$ reaches moderate amounts – i.e. when it is located between two threshold values, $\Theta_{LTD}^{start}$ and $\Theta_{LTD}^{stop}$- $W_{pre}$ drops (LTD); whereas $W_{pre}$ rises (LTP) if $y_{CB1R}$ is larger than a third threshold, $\Theta_{LTP}^{start}$ (see the dashed lines in *Figure 3C1 and 3C2* and summary in *Figure 3D*). $W_{pre}$ remains unchanged outside those ranges, i.e. if $y_{CB1R} < \Theta_{LTD}^{start}$ or if $\theta_{LTD}^{stop} < y_{CB1R} < \Theta_{LTP}^{start}$. Combining this mechanism with the shape of $y_{CB1R}$ evolution upon N$_{pairings}$ explains the main characteristics of corticostriatal STDP. With short pre-post pairings (10<Δt$_{STDP}$<40 ms), $y_{CB1R}$ reaches the LTD range (between $\Theta_{LTD}^{start}$ and $\Theta_{LTD}^{stop}$ *Figure 3C2*) during most of the 100 pairings: each pairing reduces $W_{pre}$. Since pre-post pairings do not alter $W_{post}$ (*Figure 2B2*), the net result is a progressive reduction of $W_{total}$, i.e. the expression of eCB-tLTD. The situation is different for post-pre pairings. The amplitude of $y_{CB1R}$ peaks overcomes $\Theta_{LTP}^{start}$ for 5 to 30 post-pre-pairings, resulting in an increase of $W_{pre}$. Since more than 50 post-pre pairings are needed to alter $W_{post}$ (*Figure 2B2*), this $W_{post}$ increase results in eCB-tLTP (*Figure 3C1*). Above 30 post-pre pairings, the amplitude of $y_{CB1R}$ transients gets back below $\Theta_{LTP}^{start}$ so that the $W_{pre}$ increase is no more triggered, thus explaining why eCB-tLTP is not expressed for N$_{pairings}$>30. Finally, when N$_{pairings}$>50, $W_{post}$ is predicted to trigger the rise of $W_{total}$, thus reflecting NMDAR-tLTP.

In conclusion, the mechanism proposed by our mathematical model to account for eCB-STDP is the following: eCB-tLTD requires moderate levels of CB1R activation, which can be reached with pre-post pairings; eCB-tLTP demands higher levels of CB1R activation that are reached only with 5–30 post-pre pairings, where every component of the model contributes maximally to CB1R activation (maximal cytosolic calcium influx from NMDAR, VSCC, TRPV1 and maximal calcium efflux from internal stores, combined with a minimal CB1R desensitization). Beyond 30 post-pre pairings, calcium efflux from the internal calcium stores decreases while in parallel CB1R desensitization increases. CB1R activation becomes insufficient to maintain the elevation of the synaptic weight, so that eCB-tLTP vanishes.

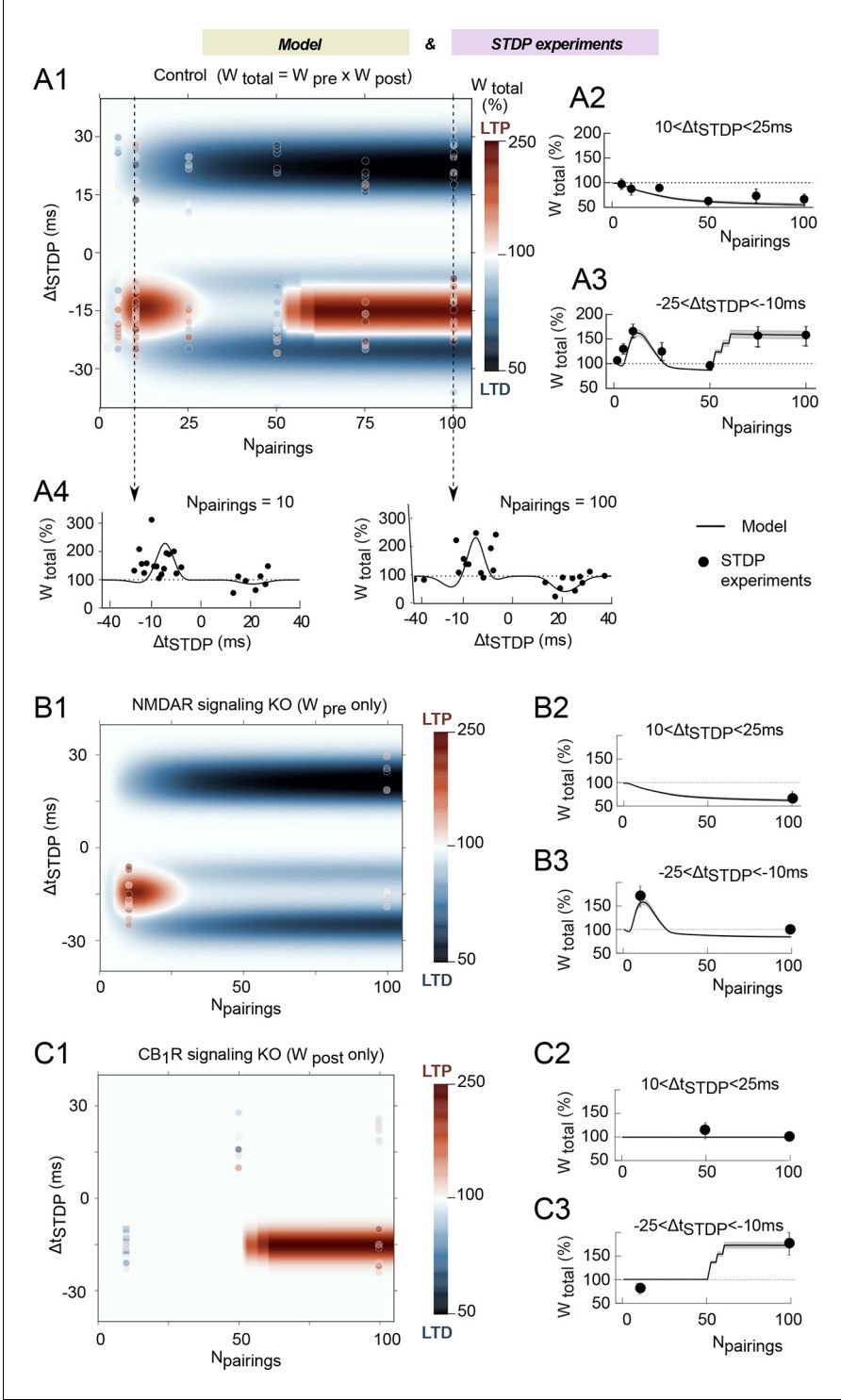

**Figure 4.** The mathematical model matches the experimental data. (**A**) Changes of the total synaptic weight $W_{total}$ (LTP and LTD) when $N_{pairings}$ and $\Delta t_{STDP}$ vary. (**A1**) Color-coded changes of $W_{total}$ in the ($N_{pairings}$, $\Delta t_{STDP}$) space. The color bar indicates the color code. The background map shows the simulation results whereas the color-coded points (same color-code as the simulations) are experimental results. The average changes with $N_{pairings}$ of $W_{total}$ integrated over short positive or short negative $\Delta t_{STDP}$ are shown in (**A2**) and (**A3**), respectively. Cross-sections of the two-dimensional map (**A1**) along the *N*-axis are shown as changes of $W_{total}$ with $\Delta t_{STDP}$, for $N_{pairings}$=10 (**A4**) or 100 (**A5**) pairings at 1 Hz. In (**A2-A5**), full black lines represent the simulation results whereas the full black circles show experimental results. (**B,C**) Corresponding results obtained with variants of the

*Figure 4 continued on next page*

*Figure 4 continued*

mathematical model where NMDAR-signaling (**B**) or eCB-signaling (**C**) were knocked-out in silico. The 2D maps (**B1, C1**) use the same color code and symbols as (**A1**). The average changes of $W_{total}$ over short positive or short negative spike timings $\Delta t_{STDP}$ (**B2,C2**) and (**B3,C3**), respectively, use the same symbols as (**A1-2**).

The following figure supplement is available for figure 4:

**Figure supplement 1.** Robustness of the model.

## The mathematical model accounts for bidirectional eCB- and NMDAR-mediated STDP

We then tested whether the model generated correct qualitative predictions in agreement with experimental data for the plasticity outcome when both $\Delta t_{STDP}$ and $N_{pairings}$ were varied. The changes of the total synaptic weight for the whole range of $\Delta t_{STDP}$ and $N_{pairings}$ are illustrated in *Figure 4A* by the model-generated color-coded map. The outcome of plasticity according to the model is split along three domains: a first LTP domain for -3<$\Delta t_{STDP}$<-25 ms and 3<$N_{pairings}$<40, a second LTP domain for -10<$\Delta t_{STDP}$<-25 ms and $N_{pairings}$>50, and a LTD domain for 10<$\Delta t_{STDP}$<25 ms and $N_{pairings}$>20. Note that the model correctly accounts for a plasticity gap for 40–60 post-pre pairings that isolates the two LTP domains in agreement with experimental observations (*Cui et al., 2015*) and that the expression of plasticity does not change when $N_{pairings}$>100 (*Figure 4—figure supplement 1A*). To compare model prediction and experimental data on a quantitative basis, *Figure 4A2 and A3* also show the average weight change predicted for -25<$\Delta t_{STDP}$<-10 ms or 10<$\Delta t_{STDP}$<25 ms. Even quantitatively, model predictions (full lines) are in agreement with the experimental data (full circles). Likewise, *Figure 4A4 and A5* show the weight change predicted for STDP protocols featuring 10 or 100 pairings, with $\Delta t_{STDP}$ ranging from -40 to 40 ms, i.e. cross-sections of the color-coded map along the vertical dashed lines. Again, model prediction (full lines) matches experiments (full circles). Quantitative agreement is found for the amplitude and sign of plasticity, as well as for the dependence of plasticity on spike timing. To our knowledge, the present model is the first mathematical model able to account for the outcome of the plasticity when both $\Delta t_{STDP}$ and $N_{pairings}$ are varied.

   We ran simulations of model variants where parts of the signaling pathways were removed (in-silico knock-out). In the NMDAR signaling knockout, we removed the whole signaling pathway downstream of NMDAR, i.e. calmodulin and CaMKII. Since $W_{post}$ relies entirely on CaMKII activation, the NMDAR signaling knockout corresponds to a situation where the contribution of $W_{post}$ is absent and only $W_{pre}$ contributes to $W_{total}$. As expected, the post-pre NDMAR-dependent LTP is absent in this NMDAR signaling knockout model, but pre-post tLTD and post-pre tLTP (observed with low numbers of pairings: 5<$N_{pairings}$<35) are conserved (*Figure 4B*). Comparison with experimental data where NMDAR signaling was blocked with D-AP5 or CaMKI with KN62 confirms the match between model and experiments (*Figure 4B*). Simulations of the CB1R in-silico knockout model, where CB1R activation remains null whatever eCB levels are shown in *Figure 4C*. Because $W_{pre}$ depends on CB1R activation, the CB1R in-silico knockout model actually reflects the case were only $W_{post}$ contributes to $W_{total}$. In this case, the only remaining plasticity domain is the LTP expressed for post-pre pairings ($N_{pairings}$>50). Again, averaging over -25<$\Delta t_{STDP}$<-10 ms and 10<$\Delta t_{STDP}$<25 ms with 10 or 100 pairings evidences the match of the model with experimental data in which CB1R was inhibited with AM251 (*Figure 4C*).

   We then analyzed how much the model outcome was sensitive to variations of the parameters. First, we changed the sharp thresholds for eCB-dependent plasticity into smooth thresholds. To this end, we replaced function $\Omega$ in *Equation 1* above by a smooth equivalent function whose graph is depicted in *Figure 4—figure supplement 1B* (the corresponding equation is given in *Supplementary file 2*, eq.S1-S2). In spite of the smooth thresholds, the model output is very similar to that obtained with sharp thresholds (compare the color map of *Figure 4—figure supplement 1B* with that of *Figure 4A1*). Therefore, our choice of a sharp thresholding for eCB-dependent plasticity is not crucial for the model output.

   We further undertook sensitivity analysis of the model (*Figure 4—figure supplement 1C*). As expected, the most sensitive parameters were those related to reactions that are known from

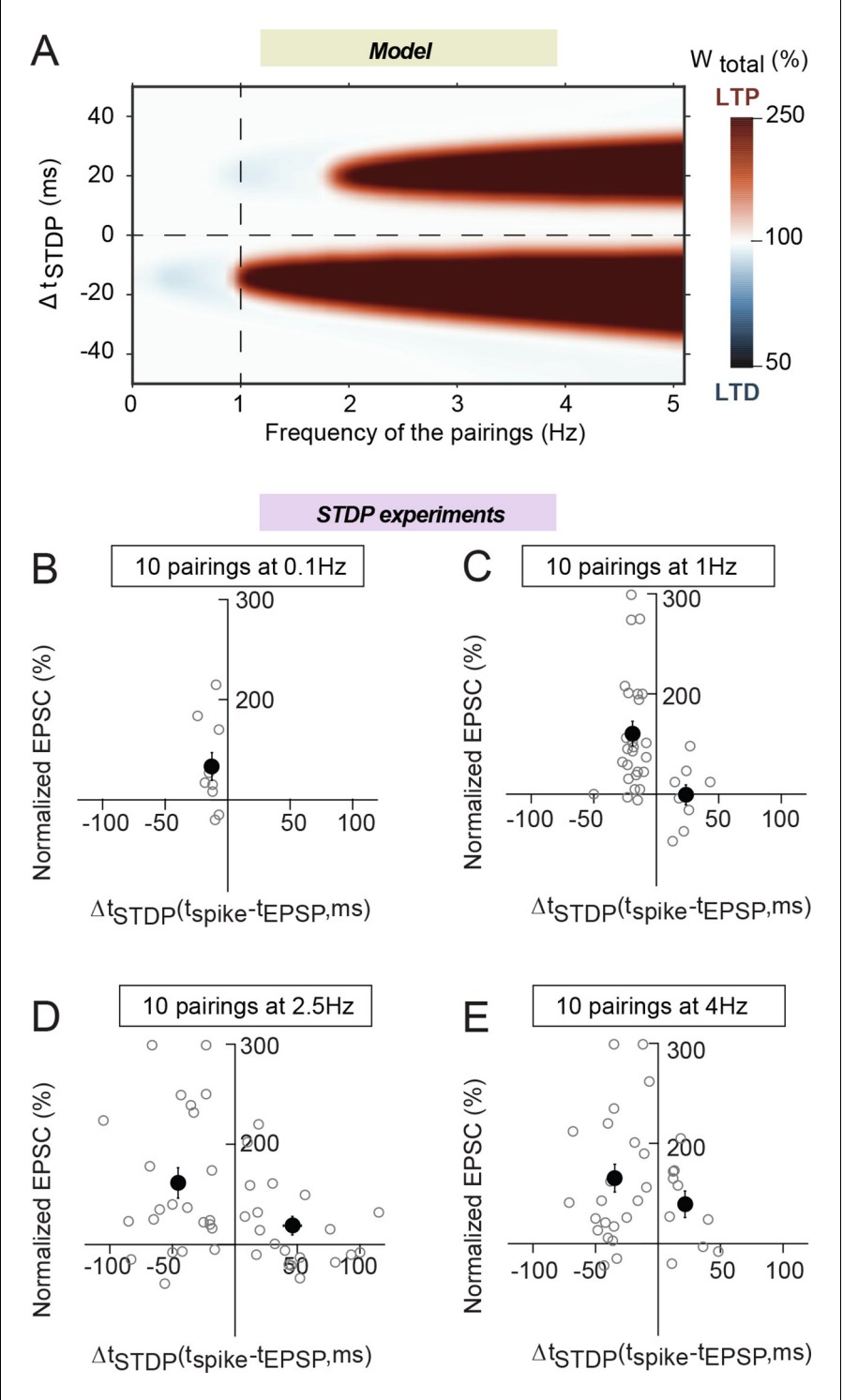

**Figure 5.** Frequency dependence of eCB-tLTP. (**A**) Color-coded changes of $W_{total}$ in the ($\Delta t_{STDP}$, frequency) parameter space for 10 pairings. Except the pairing frequency, all parameters are the same as in **Figure 4** (values given in **Supplementary file 1A-C**). For 10 post-pre pairings ($\Delta t_{STDP}<0$), tLTP disappears quickly below 1 Hz but is maintained above 1 Hz, within an even enlarged $\Delta t_{STDP}$ range. For 10 pre-post pairings ($\Delta t_{STDP}>0$), a new tLTP emerges for frequencies larger than 2 Hz. (**B-D**) Summary graphs of STDP occurrence for 10 pairings at 0.1 Hz (**B**), 1 Hz (**C**), 2.5 Hz (**D**) and 4 Hz (**E**); each grey empty circle represent the synaptic efficacy changes 45–50 min after pairings protocols for a single neuron; the black circles represent the averages of plasticity. tLTP was induced with 10 post-pre pairings at 0.1 Hz (7/10 cells showed significant tLTP) and 1 Hz (21/25 cells showed significant tLTP); no

*Figure 5 continued on next page*

*Figure 5 continued*

significant plasticity was observed for pre-post pairings. For 10 pairings at 2.5 and 4 Hz, symmetric Hebbian plasticity (tLTP for post-pre and pre-post pairings) was observed in an enlarged $\Delta t_{STDP}$; at 2.5 Hz for post-pre and pre-post pairings, 18/23 and 10/20 cells displayed significant tLTP; at 4 Hz for post-pre and pre-post pairings, 18/22 and 7/10 cells showed significant tLTP. Normality was assumed for the post-pre pairings data (test not passed).

The following figure supplement is available for figure 5:

**Figure supplement 1.** Both CB1R and NMDAR are involved in symmetric hebbian plasticity induced with 10 pairings at 4 Hz.

---

pharmacological experiments to be indeed crucial to STDP: the total amount of Calmodulin or CaM-KII (*Figure 1—figure supplement 1*), post-synaptic calcium buffering (*Fino et al., 2010*; *Cui et al., 2015*), TRPV1 and NMDA channels (*Fino et al., 2010*; *Cui et al., 2015*), DAGLipase activity (*Cui et al., 2015*) or FAAH and MAGLipase activity (see below). The model was also found sensitive to the dynamics of CB1R desensitization, in agreement with the importance of CB1R desensitization in the decay of eCB-LTP above 15–20 post-pre stimulations. The model was also sensitive to the value of the threshold for eCB-LTP induction (whether smooth or sharp). We suspect that this could explain the dispersion of the amplitudes of eCB-tLTP (*Figure 4A4*). More surprising is the sensitivity of the model to the dynamics of glutamate in the synaptic cleft (decay rate $\tau_G$). Alterations of the dynamics of glutamate release and uptake can thus be expected to play an important role in the control of STDP at the corticostriatal synapse.

## Frequency dependence of eCB-tLTP

In addition to spike timing and number of pairings, STDP is also known to be dependent on the pairing frequency. All our above results were obtained at 1 Hz. We now test the frequency dependence of plasticity induced by a low number of pairings. *Figure 5A* shows the prediction of the model for $N_{pairings}=10$. When frequency increases above 1 Hz, the eCB-tLTP triggered by post-pre stimulations ($\Delta t_{STDP}<0$) persists and is even observed for an increasingly large $\Delta t_{STDP}$ range. The model also predicts the expression of another tLTP, triggered by 10 pre-post stimulations ($\Delta t_{STDP}>0$) for frequency larger than 2 Hz.

To test the validity of these model predictions, we explored experimentally 10 pairings STDP for 0.1, 2.5 and 4 Hz (besides 1 Hz). 10 post-pre pairings at 0.1 Hz were able to induce tLTP (133±14, n=10, p=0.0386) (*Figure 5B*), which was not significantly different from eCB-tLTP induced with 10 pairings at 1 Hz (p=0.1538) (*Figure 5C*). This result is not predicted by the model, for which the tLTP induced by 10 post-pre pairings vanishes quickly below 1 Hz. At frequencies >1 Hz, we observed tLTP for 10 post-pre pairings at 2.5 Hz (161±15, n=23, p=0.0004) and 4 Hz (165±14, n=22, p=0.0001), but also for pre-post pairings at 2.5 Hz (130±14, n=12, p=0.0490 for $\Delta t_{STDP}<+50$ ms; 119±9, n=20, p=0.060 for $\Delta t_{STDP}<+100$ ms) and 4 Hz (139±13, n=10, p=0.0150). Moreover, the $\Delta t_{STDP}$ range for tLTP induction was considerably enlarged for post-pre pairings: from -30<$\Delta t_{STDP}$<0 ms at 1 Hz to -100<$\Delta t_{STDP}$<0 ms at 2.5 or 4 Hz. Note that for pre-post pairings, tLTP could be observed for $\Delta t_{STDP}<+50$ ms (*Figure 5D and E*). Therefore, when we increased the frequency of the pairings to 2.5 or 4 Hz, our experimental results show a very good match with the prediction of the model: we observed first a symmetric Hebbian plasticity, i.e. the induction of tLTP not only for post-pre but also for pre-post pairings, and, secondly an enlargement of the range of $\Delta t_{STDP}$ in which plasticity was observed.

We then investigated the signaling pathways involved in those two tLTP (*Figure 5—figure supplement 1A*). We observed that for 2.5 and 4 Hz STDP, post-pre tLTP was not prevented with AM251 (3 μM) (150±11, n=6, p=0.0069) or with D-AP5 (50 μM) (135±12, n=11, p=0.013) but was precluded with a mixture of both AM251 and D-AP5 (96±3, n=9, p=01800). Similarly, for pre-post pairings at 2.5 and 4 Hz, tLTP was still observed with AM251 (149±15, n=7, p=0.0178) but was prevented with D-AP5 (134±27, n=5, p=0.2684) or a mixture of AM251 and D-AP5 (88±11, n=3, p=0.4090). The mathematical model with $N_{pairings}=10$ does not show such a mixed NMDAR- and eCB-LTP (both tLTP are purely eCB-dependent). Remarkably, however, the tLTP in the model starts becoming mixed with $N_{pairings}>12$. For 15 pairings (at 4 Hz), for instance (*Figure 5—figure*

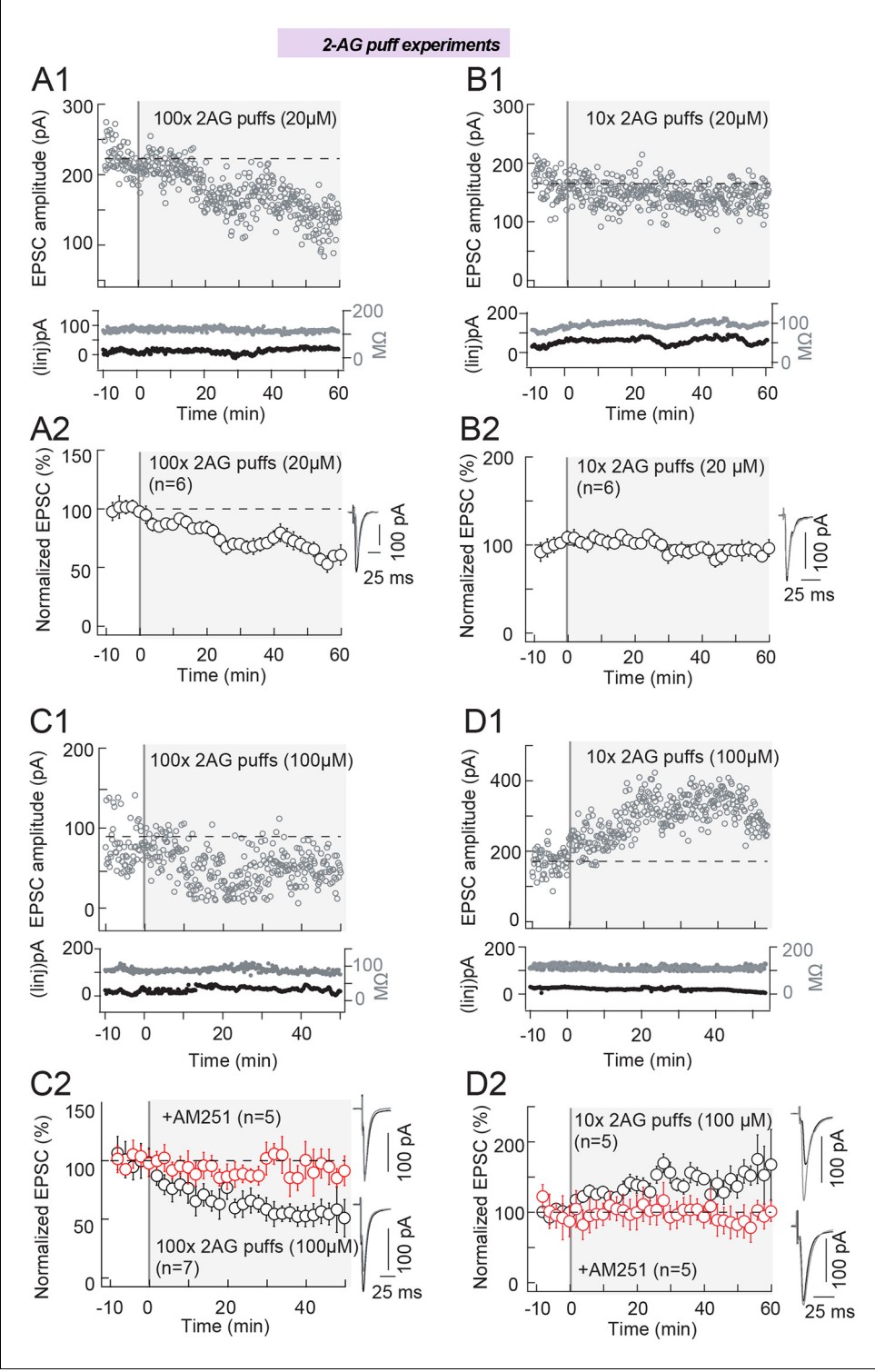

**Figure 6.** 2-AG level and duration control the eCB-plasticity polarity. (**A**) Repeated (100 times) brief application of 2-AG (at a low concentration, 20 µM) induces LTD. A series of 100 2-AG puffs (20 µM, 300 ms duration each) delivered at 1 Hz at the vicinity (50–100 µm) of the recorded striatal neuron, induced LTD in the absence of any STDP protocol (n=6). (**A1**) Example of LTD induced with 100 puffs of 20 µM 2-AG. Top, EPSC strength before and after 2-AG puffs (before 2-AG puffs: 223±3 pA; 45–55 min after 2-AG puffs: 144±3 pA; decrease of 35%). Bottom, time courses of Ri (before, 118±1 MΩ; after, 111±1 MΩ; change of 6.2%) and injected current (Iinj) for this cell. (**A2**) Summary of LTD induced with 100 puffs of 20 µM 2-AG; 6/6 cells showed significant LTD. (**B**) Repeated (10 times)
*Figure 6 continued on next page*

*Figure 6 continued*

brief application of 2-AG (at a low concentration, 20 µM) failed to induce plasticity. (**B1**) Example of absence of plasticity observed with 10 puffs of 20 µM 2-AG. Top, EPSC strength before and after 2-AG puffs (before 2-AG puffs: 165±3 pA; 45–55 min after 2-AG puffs: 143±2pA; change of 14%). Bottom, time courses of Ri (before, 82±1MΩ; after, 93±1MΩ; change of 13%) and injected current (Iinj) for this cell. (**B2**) Summary of absence of plasticity observed with 10 puffs of 20 µM 2-AG; 2/6 cells showed no significant plasticity. (**C**) Repeated (100 times) brief application of 2-AG (at a high concentration, 100 µM) induces LTD. (**C1**) Example of LTD induced with 100 puffs of 100 µM 2-AG. Top, EPSC strength before and after 2-AG puffs (before 2-AG puffs: 95±3 pA; 45–55 min after 2-AG puffs: 77±1 pA; decrease of 19%). Bottom, time courses of Ri (before, 92±1 MΩ; after, 78±1 MΩ; change of 15%) and injected current (Iinj; before, 16±1 pA; after, 26±1 pA; change of 12.8%) for this cell. (**C2**) Summary of LTD induced with 100 puffs of 100 µM 2-AG; 7/7 cells showed significant LTD. This 2-AG-mediated LTD was prevented by AM251 (3 µM, n=5); 5/5 cells showed no significant LTD. (**D**) Repeated (10 times) brief application of 2-AG (at a high concentration, 100 µM) induces LTP. (**D1**) Example of LTP induced with 10 puffs of 100 µM 2-AG. Top, EPSC strength before and after 2-AG puffs (before 2-AG puffs: 171±4 pA; 45–55 min after 2-AG puffs: 331±6 pA; increase of 94%). Bottom, time courses of Ri (before, 119±1 MΩ; after, 109±1 MΩ; change of 8.4%) and injected current (Iinj) (before, 26±1 pA; after, 18±1 pA; change of 4.7%) for this cell. (**D2**) Summary of LTP induced with 10 puffs of 100 µM 2-AG; 4/5 cells showed significant LTP. This 2-AG-mediated LTP was prevented by inhibition of CB1R with AM251 (3 µM, n=5); 5/5 cells showed no significant plasticity. Example recording monitoring EPSCs (at 0.1 Hz) (**A1**, **B1**, **C1** and **D1**) before and after 2-AG puffs, together with the time course of Ri and of the injected current (Iinj). Summary (**A2**, **B2**, **C2** and **D2**) show global average of experiments with error bars representing s.d. Representative traces are the average of 15 EPSCs during baseline (black traces) and 50 min after STDP protocol (grey traces). *p<0.05. ns: non-significant.

*supplement 1B*), the post-pre LTP in the model depends both on CB1R and NMDAR. Therefore, model predictions and experiments provide converging suggestion that at frequencies above 1 Hz, the tLTP triggered by 10–15 post-pre or pre-post pairings becomes both eCB and NMDAR-dependent.

## Level and duration of 2-AG release control the eCB-plasticity polarity

Based on the ability of our mathematical model to reproduce our experimental data, we explored further the biochemical mechanisms of eCB-dependent plasticity using a model-guided experimental strategy. Our strategy was to use the model to propose experiments that would question the role of the amplitude of CB1R activation to determine eCB-STDP polarity (LTP or LTD). We then systematically carried out the experiments necessary to test the validity of the model prediction.

We first tested experimentally the main prediction of the model: different levels of released 2-AG, low or high, would orientate the plasticity toward, respectively, eCB-tLTD or eCB-tLTP. For this purpose, we directly applied brief puffs (300 ms duration) of 2-AG (at low, 20 µM, or high, 100 µM, concentrations) either 100 or 10 times at 1 Hz, thus with the same total duration as the 100 and 10 pairings STDP protocol at 1 Hz.

First, we tested a low [2-AG] (20 µM) by delivering 100 and 10 puffs. We observed that in the absence of STDP protocol, 100 puffs of 2-AG were able to induce a significant LTD (65±5%, p=0.0009, n=6) (*Figure 6A1 and 6A2*) with magnitude similar to the tLTD induced by 100 pre-post pairings (Fig*Figure 1D2*) (p=0.9340). When we delivered 10 puffs of low [2-AG] (20 µM), no significant plasticity was detected (95±11%, p=0.6931, n=6) (*Figure 6B1 and 6B2*).

We then increased [2-AG] five-fold (i.e. 100 µM). After applying 100 puffs of 100 µM 2-AG, a potent LTD was observed (61±7%, p=0.0021, n=7) (*Figure 6C1 and 6C2*) with magnitude similar to the tLTD induced by 100 pre-post pairings (*Figure 1D2*) (p=0.6676). We verified that this LTD was CB1R-mediated by preventing plasticity with AM251 (3 µM) (94±5%, p=0.2817, n=5) (*Figure 6C2*). Strikingly, 10 puffs of high[2-AG] (100 µM) induced a potent LTP (168±29%, p=0.0106, n=5) (*Figure 6D1 and 6D2*) with magnitude similar to the tLTP induced by 10 post-pre pairings (*Figure 1E2*) (p=0.0106). This LTP was CB1R-mediated because when CB1R was inhibited with AM251 (3 µM), 10 puffs of [2-AG] (100 µM) did not induce significant plasticity (92±4%, p=0.1542, n=5) (*Figure 6D2*).

Therefore, 100 puffs of low or high [2-AG] induce LTD while only high [2-AG] succeeds to trigger LTP, thus validating the model prediction.

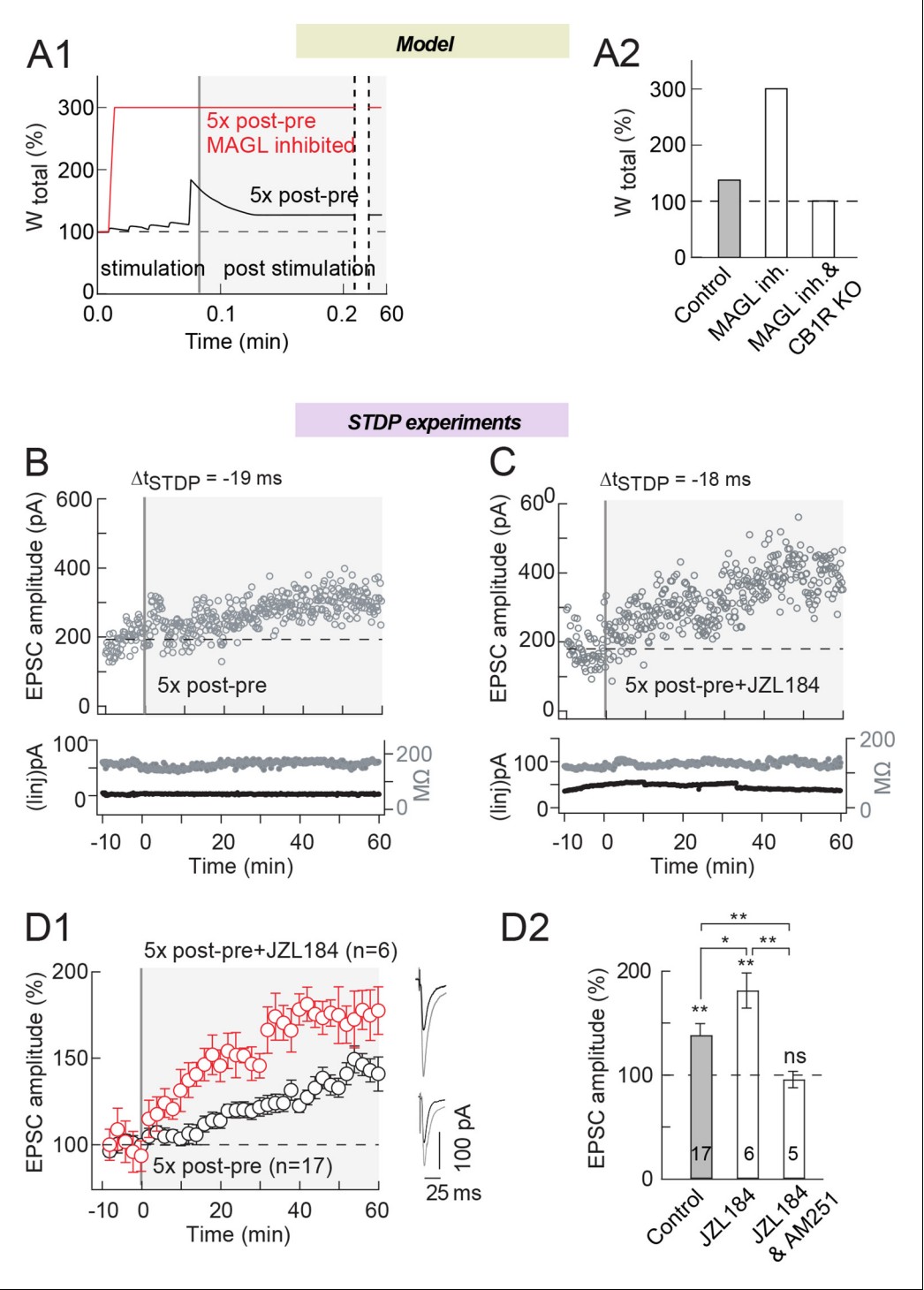

**Figure 7.** MAGL inhibition increases eCB-tLTP magnitude induced by 5 pairings. (**A**) Model prediction for eCB-LTP amplitude induced by $N_{pairings}$=5 post-pre pairings with $\Delta t_{STDP}$=-15 ms. (**A1**) In control (full black line), the synaptic weight increases during the 5 s stimulation protocol (white background) and stabilizes afterwards (gray background) to a moderate tLTP. When eCB production is amplified in the model by MAGL inhibition and DAG-Kinase activity (full red line), the amplitude of the tLTP resulting from the same stimulation is drastically amplified. In the model, MAGL and DAG-Kinase inhibition were simulated by fixing the value of the maximal rates of each enzyme to 0 and 5%, respectively, of their default values listed in *Supplementary file 1C*. (**A2**) Summary bar graph of the tLTP amplitude predicted by the model for $N_{pairings}$=5 post-pre pairings, $\Delta t_{STDP}$=-15 ms. (**B**) Corresponding example of experimental tLTP induced by 5 post-pre pairings. Top, EPSC strength before and after pairings

*Figure 7 continued on next page*

*Figure 7 continued*

(before pairings: 145±4 pA; 45–55 min after pairings: 247±5 pA; increase of 70%). Bottom, time courses of Ri (before, 165±1 MΩ; after, 170±1 MΩ; change of 3.0%) and injected current (Iinj) (before, 2±0.2 pA; after, 3±0.1 pA; change of 0.7%) for this cell. (C) MAGL inhibition by JZL184 (1.5 µM) led to an increase of tLTP magnitude. Example of tLTP induced by 10 post-pre pairings with bath-applied JZL184. Top, EPSC strength before and after pairings (before pairings: 180±6 pA; 45–55 min after pairings: 412±7 pA; increase of 129%). Bottom, time courses of Ri (before, 116±1 MΩ; after, 126±1 MΩ; change of 8.6%) and injected current (Iinj) (before, -13±1 pA; after, -8±1 pA; change of 2.8%) for this cell. (D) Summary of tLTP induced by 5 post-pre pairings in control conditions and with JZL184 treatment. 10/17 and 6/6 cells showed significant tLTP in control and in JZL184, respectively. Normality was assumed for the ctrl 5x post-pre data (test not passed). (E) Summary bar graph illustrates that tLTP magnitude was increased by MAGL inhibition (JZL184) while prevented by CB1R inhibition (JZL184 1.5 µM +AM251 3 µM). Representative traces are the average of 15 EPSCs during baseline (black traces) and 50 min after STDP protocol (grey traces). *p<0.05. ns: non-significant.

## Alterations of MAG-lipase activity evidence the key role of 2-AG concentration in gating eCB bidirectional plasticity

To further substantiate the causal role of the amplitude of 2-AG transients in bidirectional eCB-plasticity, we boosted the endogenous levels of 2-AG during STDP protocols. Indeed, if the amplitude of CB1R activation controls the expression of eCB-STDP, the outcome of a given STDP protocol should change if one modifies the amount of CB1R activated by this very same STDP protocol. For this purpose, we inhibited the MAG lipase (MAGL), the major enzyme responsible for 2-AG degradation (*Piomelli, 2003*), to increase the endogenous level of 2-AG.

We took advantage of the model to select three scenarios in which it should be possible in silico, by inhibiting MAGL, to 1) increase the magnitude of an existing eCB-tLTP, 2) induce an eCB-tLTP for a paradigm which normally exhibits neither eCB-tLTP nor NMDAR-LTP (i.e. 50 post-pre pairings; *Figure 4A* and *Cui et al., 2015*) and 3) convert an eCB-LTD (induced with 100 presynaptic stimulations without postsynaptic simulations) into eCB-LTP.

First, we tested the possibility to increase the eCB-LTP magnitude by inhibiting MAGL. For this purpose, we chose the minimal pairing protocol for which we detected eCB-LTP, which is five post-pre pairings (*Figure 7*) (*Figure 4A3* and see *Figure 6* in *Cui et al., 2015*). 5Five pairings appearas the lowest number of pairings needed to induce significant eCB-tLTP as illustrated by the representative and average STDP (134±13%, p=0.0190, n=17) (*Figure 7B and D*); Note that the model also faithfully predicted eCB-tLTP for such number of pairings (*Figure 4A* and *7A*). In the model, we introduced noncompetitive inhibition of the MAGL by decreasing its maximal rate $r_{MAGL}$ (*Supplementary file 1C*). Simulation of the model with 5 post-pre pairings under MAGL inhibition predicts that such an inhibition increases the net level of 2-AG produced during the protocol and the amplitude of eCB-LTP (*Figure 7A*). As predicted by the model, inhibition of the MAGL with JZL184 (1.5 µM) significantly increased the magnitude of eCB-tLTP (182±17%, p=0.0048, n=6; p=0.0294 when compared to 5 post-pre pairings in control conditions) (*Figure 7*: with an example of LTP induced by five 5 post-pre pairings at $\Delta t_{STDP}$=-19 ms in B, with an example of LTP induced by five post-pre pairings at $\Delta t_{STDP}$=-18 ms with JZL184 in C and the experiment summary in D). We confirmed that this amplification was CB1R-mediated since no plasticity was observed when CB1R were blocked by AM251 (3 µM) (96±8%, p=0.6123, n=5) (*Figure 7D*). We also ensured that bath-applied JZL treatment in the absence of STDP pairings did not induce significant plasticity (96±7%, p=0.5943, n=5). It should be noted that the occurrence of eCB-tLTP was also higher with MAGL inhibition: in control, five post-pre pairings yielded 60% of eCB-tLTP (10/17 cells showed significant tLTP) while with MAGL inhibition, 100% of the recorded cells displayed eCB-tLTP (6/6 cells displayed significant tLTP).

Second, we tested the possibility to induce eCB-tLTP by inhibiting MAGL. Indeed, our model predicts that MAGL inhibition may turn a STDP protocol that yields no plastic change in control conditions into eCB-tLTP. For this purpose, we chose a STDP pairing for which we detected no plasticity in control conditions: i.e. 50 post-pre pairings (*Figure 4A3*; *Cui et al., 2015*). *Figure 4A3* and *Figure 8* illustrates this 'plasticity gap' (the zone between 40 and 60 pre-post pairings that separates the two LTP domains). In silico the control STDP protocol (50 pairings with $\Delta t_{STDP}$=-15 ms) does not trigger any plasticity but when MAG lipase is inhibited, eCB-tLTP emerges (*Figure 8A*).

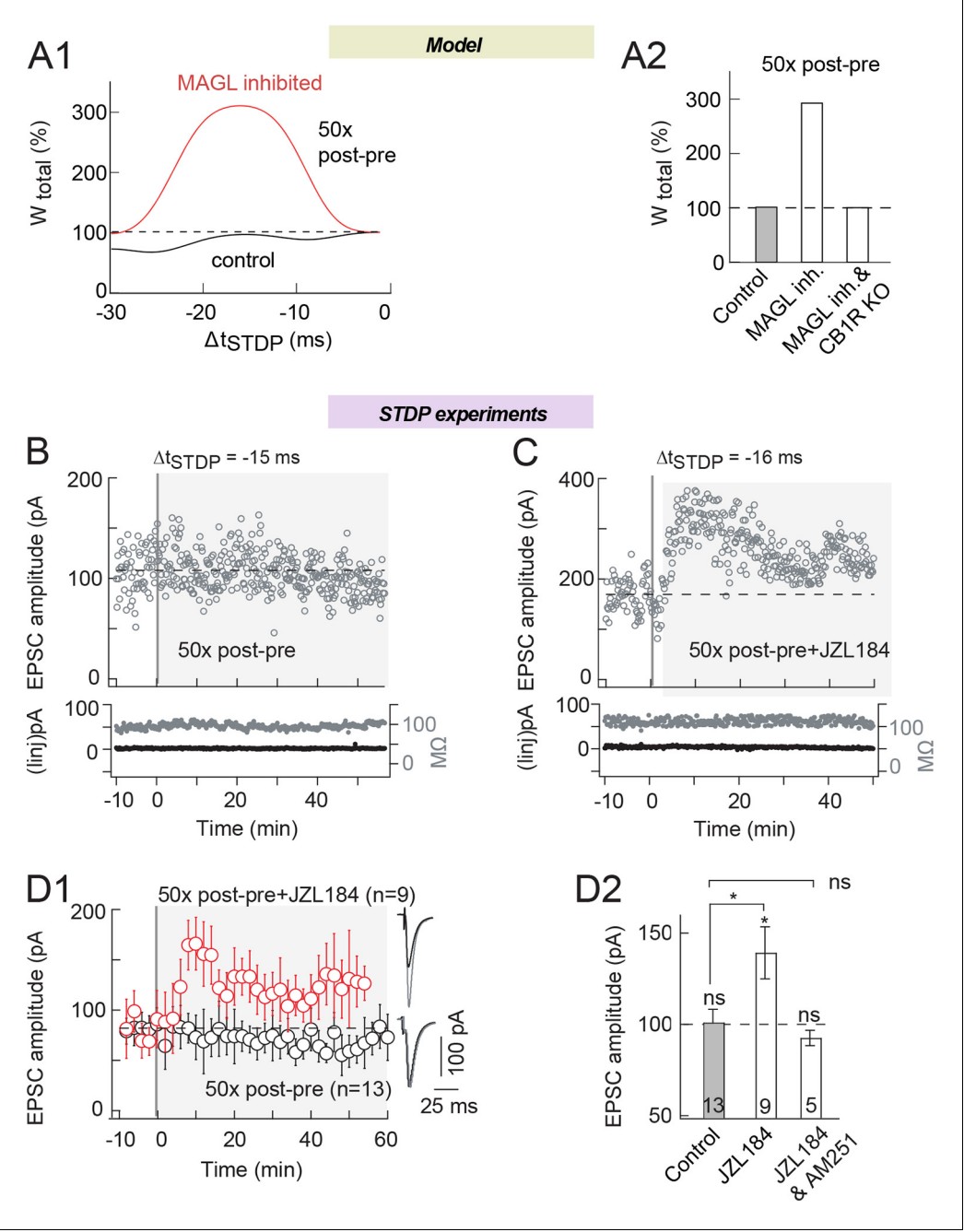

**Figure 8.** MAGL inhibition unveils eCB-tLTP expression with 50 pairings. (**A**) Model prediction for the plasticity induced by $N_{pairings}$=50 post-pre pairings. (**A1**) In control (full black line), the synaptic weight is unchanged by 50 post-pre pairings for $0>\Delta t_{STDP}>-25$ ms. Amplified eCB production due to MAGL inhibition (full red line), uncovers a large-amplitude tLTP. In the model, MAGL inhibition was emulated by setting its maximal rate to 40% of its default value (***Supplementary file 1C***). (**A2**) Summary bar graph of the tLTP amplitude predicted by the model for $N_{pairings}$=50 post-pre pairings at $\Delta t_{STDP}$ =-15 ms. (**B**) 50 post-pre pairings did not induce significant plasticity. Example of the absence of plasticity observed when 50 post-pre pairings were applied. Top, EPSC strength before and after pairings (before pairings: 108±3 pA; 45–55 min after pairings: 106±2; change of 2%). Bottom, time courses of Ri (before, 92±1 MΩ; after, 91±1 MΩ; change of 0.1%) and injected current (Iinj) (before, 2±0.2 pA; after, 2±0.1 pA; no detectable change) for this cell. (**C**) 50 post-pre pairings induced tLTP with MAGL inhibition. Example of tLTP induced by 50 post-pre pairings with bath-applied JZL184 (1.5 µM). Top, EPSC strength before and after pairings (before pairings: 170±4 pA; 45–55 min after pairings: 243±4 pA; increase of 43%). Bottom, time courses of Ri (before, 110±1 MΩ; after, 113±1 MΩ; change of 2.7%) and injected current (Iinj) (before, 4±0.3 pA; after,

*Figure 8 continued on next page*

*Figure 8 continued*

3±0.2 pA; change of 0.6%) for this cell. (D) Summary of synaptic weight along time induced by 50 post-pre pairings in control conditions and with JZL184 treatment. 4/13 and 8/9 cells showed significant tLTP in control and in JZL184, respectively. (E) Summary bar graph illustrates that MAGL inhibition allowed tLTP to be expressed, which was CB1R-mediated since prevented by AM251 (3 µM). Representative traces are the average of 15 EPSCs during baseline (black traces) and 50 min after STDP protocol (grey traces). *p<0.05. ns: non-significant.

Experimentally, as previously reported (*Cui et al., 2015*), STDP protocols with 50 post-pre pairings failed to induce any plasticity in control conditions as illustrated by the representative and average STDP (101±7%, p=0.9030, n=13) (*Figure 8B and D*; with an example of an absence of plasticity for 50 post-pre pairings at $\Delta t_{STDP}$=-20 ms in B and the experiment summary in D). As predicted by the model, we found that 50 post-pre pairings under inhibition of MAGL with JZL184 (1.5 µM) induced tLTP (139±15%, p=0.0248, n=9) (*Figure 8C and D*; with an example of tLTP induced by 50 post-pre pairings at $\Delta t_{STDP}$=-16 ms with JZL184 and the experiment summary in D). This tLTP was eCB-mediated since suppressed by AM251 (3 µM) (93±4%, p=0.3365, n=5) (*Figure 8D2*). Therefore, by acting on the 2-AG levels, we were able to trigger eCB-tLTP for an activity pattern, which does not generate LTP in control conditions.

Our third model prediction is that amplifying 2-AG production during STDP may even eliminate the need for a coincidence between presynaptic and postsynaptic activity to express eCB-LTP. In silico, pre-post pairing coincidence is needed for the model to express plasticity. Indeed, a protocol with 100 presynaptic stimulations only (i.e. in the absence of postsynaptic stimulation), does not change $W_{total}$ in the model (*Figure 9A*). However, if we decrease the maximal rates of MAGL and DAG kinase activity (the major source of DAG consumption in the model), we obtain a robust eCB-tLTP, even in the absence of any postsynaptic stimulation. Experimentally, 100 presynaptic stimulations (without postsynaptic pairing) induced LTD (76±9%, p=0.0337, n=8), which was CB1R-mediated since prevented with AM251 (3 µM) (102±7%, p=0.8108, n=4) (*Figure 9B*; with an example of LTD induced by 100 pre stimulations in B1 and the experiment summary in B2); note that this LTD was not predicted by the model. In agreement with the model, when 2-AG levels were amplified by MAGL inhibition with JZL184 (1.5 µM), 100 pre-synaptic stimulations triggered LTP (143±17%, p=0.0299, n=11) instead of LTD in control conditions (*Figure 9C*; with an example of tLTP induced by 100 pre stimulations with JZL184 in C1 and the experiment summary in 9C2). This tLTP was eCB-mediated since it was prevented when JZL184 was co-applied with AM251 (3 µM) (93±4%, p=0.1509, n=5) (*Figure 9C2*).

To summarize, manipulating the activity of the MAGL was sufficient to (1) control the magnitude of eCB-tLTP, (2) induce eCB-tLTP or (3) even to reverse eCB-LTD into eCB-tLTP. These experimental validations of the model predictions thus support our model hypothesis that 2-AG levels control eCB plasticity in a bidirectional way, with large 2-AG levels yielding eCB-tLTP and lower levels eCB-tLTD.

## eCB-LTP maintenance relies on presynaptic PKA/calcineurin activity

We next aimed at identifying which molecular actors are responsible for the modification of the presynaptic weight that is controlled by CB1R activation. In previous reports of eCB-dependent plasticity, $W_{pre}$ was found to rely on the phosphorylation state of a yet unknown target protein involved in glutamate exocytosis, controlled by PKA and calcineurin (CaN) (*Heifets and Castillo, 2009*). In particular, PKA and CaN inhibition upon CB1R activation is thought to be involved in eCB-LTD induced with high-frequency stimulation protocol (*Heifets and Castillo, 2009*).

We thus first tested the implication of PKA and CaN in eCB-tLTD. Inhibition of PKA by bath-applied inhibitor KT5720 (1 µM) during a STDP protocol that triggers eCB-tLTD in control condition (100 pre-post pairings) did not affect the expression of eCB-tLTD (74±10%, p=0.020, n=5; p=0.8752 compared to control tLTD) (*Figure 10A*). We then tested the involvement of the phosphatase CaN activity in eCB-tLTD expression. We found that CaN inhibition by cyclosporin A (1 µM) prevented eCB-tLTD (122±18%, p=0.2560, n=6) (*Figure 10A2 and 10B*) Note that cyclosporin A being cell-permeant, we cannot distinguish from those results the location (pre- or post-synaptic) of the implicated CaN.

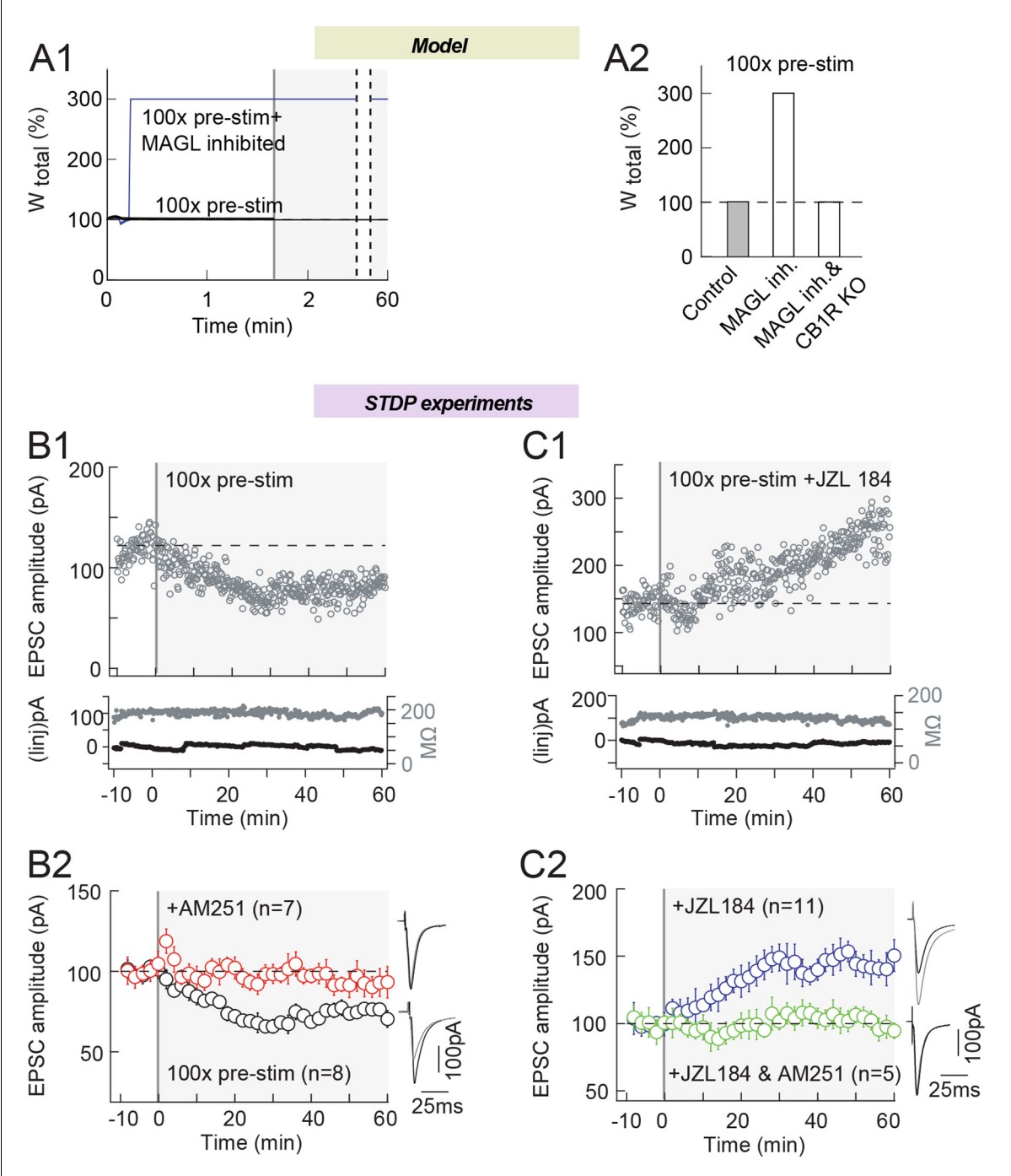

**Figure 9.** MAGL inhibition shifts eCB-LTD into eCB-LTP, induced by 100 presynaptic stimulations. (**A**) Model prediction for the changes in synaptic weight induced by 100 presynaptic stimulations. (**A1**) In the absence of postsynaptic activity, 100 presynaptic stimulations in the model do not change the synaptic weight in control conditions (full black line), but MAGL inhibition and DAG-Kinase activity (full red line) generates eCB amounts that are large enough to trigger LTP. MAGL and DAG-Kinase inhibition were obtained by fixing the value of the maximal rates of each enzyme to 0 and 5%, respectively, of their default values listed in **Supplementary file 1C**. (**A2**) Summary bar graph of the LTP amplitude predicted by the model for 100 presynaptic stimulations in the absence of postsynaptic stimulations. (**B**) Experimentally, 100 presynaptic stimulations alone (i.e. without paired postsynaptic stimulation) induced significant LTD. (**B1**) Example of LTD induced by 100 presynaptic stimulations. Top, EPSC strength before and after pairings (before pairings: 118±2 pA; 45–55 min after pairings: 78±1 pA; decrease of 34%). Bottom, time courses of Ri (before, 187±1 MΩ; after, 189±1 MΩ; change of 1.1%) and injected current (Iinj) (before, 2±0.6 pA; after, 3±0.4 pA; change of 0.8%) for this cell. (**B2**) Summary of LTD induced with 100 presynaptic stimulations; 6/8 cells showed significant LTD. This 2-AG-mediated LTD was prevented by AM251 (3 µM, n=7); 6/7 cells showed no significant plasticity. (**C**) MAGL inhibition by ZZL184 (1.5 µM) shifts eCB-LTD, induced by 100 presynaptic stimulations, into eCB-tLTP. (**C1**) Example of

*Figure 9 continued on next page*

*Figure 9 continued*

LTP induced by 100 presynaptic stimulations with MAGL inhibition. Top, EPSC strength before and after pairings (before pairings: 143±2 pA; 45–55 min after pairings: 224±2 pA; increase of 57%). Bottom, time courses of Ri (before, 131±1 MΩ; after, 131±2 MΩ; no significant change) and injected current (Iinj) (before, -3±1 pA; after, -12±1 pA; change of 6.3%) for this cell. (C2) Summary of LTP induced with 100 presynaptic stimulations; 7/11 cells showed significant LTP. This 2-AG-mediated LTD was prevented by AM251 (3 μM, n=5); 5/5 cells showed no significant plasticity. Representative traces are the average of 15 EPSCs during baseline (black traces) and 50 min after STDP protocol (grey traces). *p<0.05. ns: non-significant.

We then tested the involvement of PKA and CaN on eCB-tLTP induced with 10 post-pre pairings. Bath-applied CaN inhibitor cyclosporin A did not preclude the expression of eCB-tLTP (154±17%, p=0.0305, n=5) (*Figure 10C*). Inhibition of PKA with KT5720 (1 μM) prevented plasticity with 10 post-pre pairings (98±2%, p=0.3203, n=4) (*Figure 10C*) demonstrating that PKA activity is critically involved in eCB-tLTP. We then aimed at determining which pools of PKA (pre- and/or postsynaptic) were involved in eCB-tLTP. For this purpose, we restricted PKA inhibition to the postsynaptic neuron with intracellular application (through the patch-clamp pipette) of KT5720 (i-KT5720, 1 μM) or a cell-non-permeant PKA inhibitor PKI 6–22 (i-PKI 6–22, 20 μM). Both treatments did not significantly affect eCB-tLTP (with i-KT5720: 137±8%, p=0.0108, n=5; with i-PKI 6–22: 163±29%, p=0.03249, n=7) (*Figure 10C*). We therefore conclude that the activity of presynaptic PKA is critical for the expression of eCB-tLTP.

Together, these results suggest that the expression of eCB-tLTD at the corticostriatal synapse depends on the activity of CaN (*Figure 10B*) (and possibly on PKA inhibition), whereas the expression of eCB-tLTP is conditioned by the activity of presynaptic PKA (*Figure 10D*). Therefore, intermediate levels of CB1R activation trigger eCB-tLTD through a combination of PKA inhibition and CaN activity, whereas high levels of CB1R activation leads to eCB-tLTP through the reverse combination: PKA activity combined to CaN inhibition.

## Discussion

Long-term synaptic changes at corticostriatal synapses provide fundamental mechanisms for the function of the basal ganglia in action selection and in procedural learning (*Yin et al., 2009*) in which eCB plasticity have emerged as the major form underlying long-term synaptic strength changes (*Mathur et al., 2012*). We describe here a paired-activity dependent tLTP and tLTD, wherein eCB dynamics tightly control both the induction/maintenance and polarity of synaptic weight changes. Due to their on-demand intercellular signaling *modus operandi* (*Alger and Kim, 2011*), eCB biosynthesis and release are evoked by precisely timed physiological stimuli. Our study demonstrates that STDP, an important physiological form of Hebbian plasticity, efficiently triggers eCB signaling and that eCB signaling controls the STDP polarity in a bidirectional manner depending on the activity pattern.

Since its discovery, STDP has been attracting a lot of interest in computational neuroscience because it is based on the patterns of spike timing. Computational models of STDP can be clustered into two families. A first group of models aims at predicting the consequences of STDP on e.g. neuronal receptive fields or network dynamics (*Clopath et al., 2010*; *Costa et al., 2015*). In those models, the function describing weight changes with spike timing is usually given as hypothesis of the model. A second group of models starts from the signaling pathways implied in STDP and aims at understanding how the function describing weight changes with spike timing emerges from those signaling pathways (see e.g. *Graupner and Brunel, 2010* for a review). In a number of models in this second group, intracellular signaling is actually restricted to cytoplasmic calcium variation, thus implementing calcium-control hypothesis (*Shouval et al., 2002*). The mathematical models that consider signaling downstream of calcium usually account for a single intracellular signaling pathway (i.e. a single coincidence detector), most often NMDAR-CAMKII (*Rubin et al, 2005*, *Graupner and Brunel, 2007*; *Urakubo et al., 2008*). Noticeable exceptions are for instance *Karmarkar and Buonomano (2002)*, *Evans et al., 2012* or *Paillé et al. (2013)*, where the calcium pool entering via NMDAR controls tLTP whereas the calcium pool entering though VSCCs controls tLTD, thus implementing two coincidence detectors. However those models do not consider the signaling pathways beyond calcium entry through NMDAR and VSCC. Our mathematical model belongs to the latter

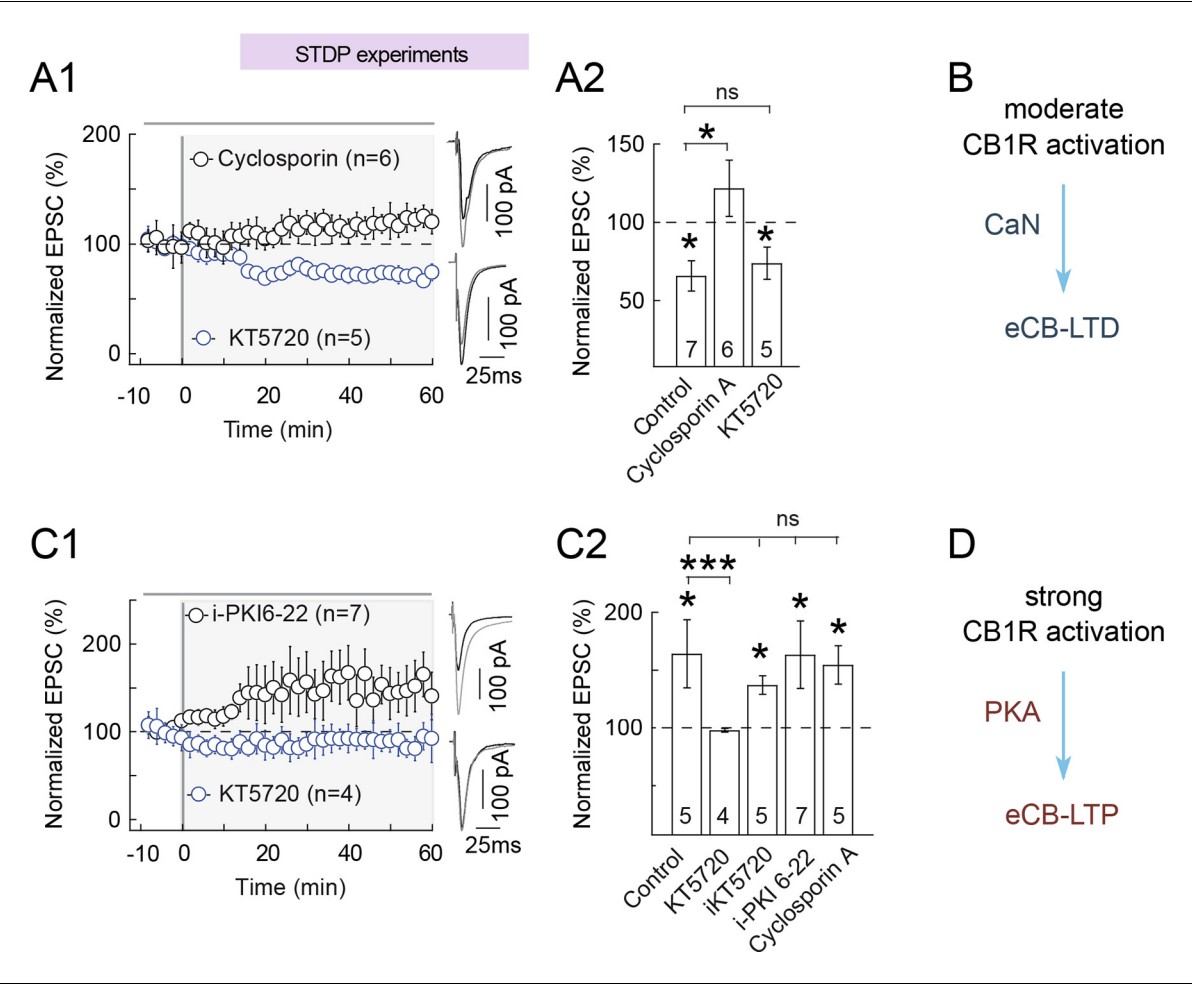

**Figure 10.** eCB-tLTP maintenance relies on presynaptic PKA activation. (A-B) eCB-tLTD is CaN-dependent. (A1) Summary of plasticity induced by 100 pre-post pairings with bath applied KT5720 (1 μM) or with cyclosporin (1 μM). eCB-tLTD was prevented with CaN (cyclosporin A) inhibition but unaffected with PKA inhibition (KT5720); 0/6 and 4/5 cells showed LTD with cyclosporin A and KT5720, respectively. (A2) Summary bar graphs illustrate that eCB-LTD maintenance involves the activation of presynaptic CaN by a CB1R-triggered signal and was independent of the activation of presynaptic PKA. (B) Main conclusion scheme: eCB-LTD is triggered by moderate prolonged levels of CB1 activations and requires active CaN. Normality was assumed for the cyclosporine A & KT5720 data (test not passed). (C-D) eCB-tLTP is PKA-dependent. (C1) Summary of plasticity induced by 10 post-pre pairings with bath applied KT5720 (1 μM) or with i-PKI6-22 (20 μM). eCB-tLTP was prevented with bath-applied KT5720 but unaffected with i-PKI6-22, a cell-impermeant PKA inhibitor applied intracellularly in the postsynaptic neuron; 0/4 and 6/7 cells showed LTP with KT5720 and i-PKI6-22, respectively. (C2) Summary bar graphs illustrate that eCB-tLTP depends on presynaptic PKA activation since it was prevented by bath-applied KT5720 but unaffected when KT5720 or PKI6-22, was applied intracellularly in the postsynaptic neuron (i-KT5720, i-PKI6-22). Cyclosporin A had no effect on eCB-tLTP, showing that it was CaN-independent. Thus, eCB-tLTP maintenance involves the activation of presynaptic PKA by a CB1R-triggered signal. Normality was assumed for the ctrl & KT5720 data (test not passed). (D) Main conclusion scheme: eCB-LTP is triggered by large levels of short duration of CB1R activation and requires presynaptic active PKA. Representative traces are the average of 15 EPSCs during baseline (black traces) and 50 min after STDP protocol (grey traces). Error bars represent SD. *p<0.05. ns: not significant.

The following figure supplement is available for figure 10:

**Figure supplement 1.** Schematic of a possible mechanism for PKA-CaN control of eCB-dependent bidirectional STDP.

group. To our knowledge, this is the first model to incorporate two detailed signaling pathways to account for STDP: NMDAR-CAMKII (with calmodulin, PKA, CaN and PP1) for tLTP and the eCB system for tLTD and tLTP. In the model, the eCB system comprises mGluR5, PLCβ, DAGL, MAGL, DAG-Kinase, calcium-induced calcium release (IP3R channels, SERCA pumps), IP3 dynamics (PLCδ, PI3K), VSCC, TRPV1R and CB1R (*Figure 2A*). Thank to this very fine grain description, the present mathematical model is able to predict the weight change when any of the STDP parameters is

varied, that is, not only spike timing $\Delta t_{STDP}$, but also $N_{pairings}$ and frequency. This capacity has allowed us to explore a novel form of plasticity, eCB-LTP, induced by a low number of post-pre pairings at 1 Hz (*Cui et al., 2015*; the present study).

Our mathematical model features 36 ordinary differential equations and roughly 150 parameters, among which more than one half is constrained by experimental data. A classical view assesses that with enough parameters, one can fit any data set, whatever the equations used. This view however does not apply in our case because we constrained the model with experimental data embedded in a three-dimensional parameter space ($\Delta t_{STDP}$, $N_{pairings}$ and frequency). In these conditions of strong constraining by experimental data, our parameter estimations systematically converged to a unique scenario that features two fundamentals: (*i*) eCB-transients allow bidirectional eCB plasticity whereby tLTD is triggered by moderate levels of eCB while high-amplitude eCB transients yield LTP and (*ii*) large eCB-transients are obtained for low numbers of pairings (for $5<N_{pairings}<20$ at 1 Hz) while for larger number of pairings ($N_{pairings}>40$ pairings at 1 Hz) reduced calcium influx from internal stores and/or CB1R desensitization curtails the amplitude of the eCB-transients. We confirmed this prediction experimentally by two means: eCB puffs with different concentrations and durations, and MAGL inhibition in various conditions. Highly concentrated short puffs of 2-AG indeed yielded LTP in the absence of any electrical stimulation whereas less concentrated prolonged 2-AG puffs produced LTD. By decreasing MAGL activity, thus favoring high 2-AG levels, we could increase the magnitude of eCB-tLTP (5 post-pre pairings), induce eCB-tLTP at 50 post-pre pairings or even switch eCB-LTD to eCB-tLTP (100 presynaptic stimulations). Therefore, under MAGL inhibition, the temporal coincidence between pre- and postsynaptic spikes is not mandatory anymore for the induction of long-term plasticity. This discovery may have far-reaching consequences, since it means that the manipulation of MAGL activity may bootstrap synaptic plasticity in synapses where the postsynaptic neuron is silent, thus rescuing possible pathological situations or waking up new local circuits. Further work is however needed to check the realism of this implication in vivo but it is noteworthy that recently, nerve growth factor (NGF) signaling in cholinergic projection neurons of fetal mice has been shown to control MAGL degradation in vivo and in vitro in a spatially specific way (*Keimpema et al., 2013*). In light of our findings, these results suggest that the regulation of MAGL activity may indeed be a potent mechanism to control synaptic plasticity and thus learning and memory.

Our experimental results showing that short puffs of highly concentrated 2-AG yields LTP in the absence of any electrical stimulation is a strong argument in favor of the model-derived hypothesis that eCB can support tLTP in addition to tLTD. Both the amplitude ($\simeq165\%$) and the pharmacology (suppression by CB1R antagonist AM251) of the LTP observed with these puffs are identical to those observed for tLTP triggered by 10 post-pre pairings. Moreover, it is important to note that when we applied 2-AG in the form of prolonged puffs (x100), we observed LTD instead of LTP. According to our mathematical model, this behavior would be due to desensitization of the CB1R that becomes prominent with prolonged puffs. 2-AG puff application is therefore generally expected to give rise to depression except if the puffs are of short duration, where potentiation can be observed. This feature may account for the widespread observation that prolonged applications of 2-AG systematically yielded to LTD and might explain why eCB-dependent potentiation has proven difficult to observe experimentally.

Our mathematical model in general shows very good agreement with experimental data, even for conditions for which the model was not fitted (stimulation frequency >1 Hz, alteration of MAGLipase activity). Some mismatches are however notable. Our focus here has mostly been on the signaling part of the system. In comparison, our modeling of the synaptic machinery and the electrophysiological response of the MSN neurons have been less sophisticated. For instance, *Figure 9B* shows that 100 presynaptic stimulations at 1 Hz, in the absence of post-synaptic stimulation, are sufficient to trigger eCB-LTD experimentally. This is likely due to the production of eCB resulting from the post-synaptic depolarization triggered by the 100 presynaptic stimulations (depolarization-induced LTD). In the model however, the postsynaptic depolarization triggered by 100 presynaptic stimulations at 1 Hz is not large enough to allow the entry of the minimal amount of calcium that is needed to over-come the eCB-LTD threshold, so no plasticity is observed.

The dynamics of glutamate release and binding are expected to be crucial for the expression of STDP, as illustrated by our sensitivity analysis (*Figure 4*). Several parts of the glutamate release and binding machinery at the corticostriatal synapse are known to display nontrivial frequency-

dependence, including presynaptic glutamate release, uptake by transporters and receptor activation because of desensitization of AMPAR (*Goubard et al., 2011*). Since our mathematical model features none of these frequency dependencies, we cannot expect a precise quantitative match between experiments and model when frequency is varied. However, it is remarkable that the model still yields correct predictions of the main qualitative trends observed in the experiments. For instance, 10 post-pre pairings at very low frequency (0.1 Hz) are sufficient to trigger tLTP in the experiments, while the model predicts no plasticity at those frequencies (*Figure 5*). At 4 Hz, the experimental pharmacology profile with 10 pairings is not matched by the model with 10 pairings, but with 15 pairings (*Figure 5*). Nevertheless, the model predictions that increased frequency should witness *i)* an enlarged range for the expression of tLTP with post-pre 10 pairings and *ii)* the emergence of a new tLTP with 10 pre-post pairings turned out to be generally correct.

In the hippocampus, theta-burst stimulation induces eCB-dependent LTD at the synapse between inhibitory interneurons and pyramidal cells that is blocked when presynaptic CaN is inhibited (*Heifets et al., 2008*). Similarly, we found that eCB-tLTD triggered at the corticostriatal synapse also needs CaN activity. In our experiments, we could not estimate the localization of the CaN that mediated eCB-LTD (pre or postsynaptic) but it is likely that in analogy with theta-burst STDP in the hippocampus, the implied CaN would be presynaptic. The implication of PKA in eCB-LTD is less clear. In the hippocampus (*Chevaleyre et al, 2007*) and nucleus accumbens (*Mato et al., 2008*), the expression of frequency-dependent eCB-LTD is blocked when cAMP levels are increased by the adenylyl-cyclase activator forskolin. Since PKA is activated by cAMP, this indicates that reduced levels of PKA are necessary for eCB-LTD. In both cases however, direct inhibition of presynaptic PKA actually blocks the expression of eCB-LTD, thus showing that the implication of PKA in eCB-dependent plasticity is a complex and subtle phenomenon. In the case of tLTD in the dorsolateral striatum, we found that direct PKA inhibition does not obliterate eCB-tLTD. Our result is, therefore, in line with the notion that the expression of eCB-tLTD requires PKA inhibition. Strikingly, the eCB-tLTP triggered at the corticostriatal synapse by STDP protocols had the exact inverse dependence on PKA and CaN compared to eCB-tLTD. Indeed, we found that PKA inhibitors block eCB-tLTP expression whereas CaN inhibition has no effect. Therefore, our study of STDP protocols at the corticostriatal synapse shows that the expression of eCB-tLTD needs CaN (but not PKA) activity whereas the expression of eCB-tLTP demands PKA (but not CaN) activity. Whether the same or two separate pathways support eCB-tLTD and eCB-tLTP is still a pending question which resolution is rendered highly challenging by the difficult access to the molecular mechanisms occurring in the presynaptic compartment of the corticostriatal synapses. However, we can propose a simplified schematic mechanism (*Figure 10—figure supplement 1*). CB1R activation inhibits PKA activity but also inhibits presynaptic VSCCs (*Mato et al., 2008*), which are expected to hamper calcium influx in the presynaptic compartment. Such a decrease in calcium levels reduces the CaN activity. CB1R activation is expected to reduce both CaN and PKA activity, although the shapes (kinetics parameters) of the decays of PKA and CaN activity with increasing CB1R activation are not necessarily identical. For example, in *Figure 10—figure supplement 1*, PKA activity dominates CaN when CB1R activation is large, whereas CaN dominates for a range of intermediate CB1R activations. Considering our experimental data, this putative mechanism suggests that eCB-STDP is gated by the ratio between PKA and CaN activities: large CB1R activation would produce high values of the PKA/CaN ratio yielding eCB-tLTP whereas intermediate CB1R activations would result in low values of the PKA/CaN ratio and, consequently, in eCB-tLTD. Future experimental investigation is needed to test the validity of this proposed mechanism.

The bidirectionality of synaptic plasticity is a key parameter since it allows LTP and LTD to reverse each another with time at a single synapse (probably at the same presynaptic side), thus enabling adaptive changes of the synaptic weight. Altogether, our results show that eCB bidirectional plasticity constitutes a versatile system, which operation may underlie a complex repertoire of learning abilities, depending on activity pattern at corticostriatal circuits and on the behavioral context.

## Materials and methods

### Ex vivo electrophysiological recordings

Brain slice preparation

All experiments were performed in accordance with local animal welfare committee (Center for Inter-disciplinary Research in Biology and EU guidelines, directive 2010/63/EU). Sprague-Dawley rats (Charles River, L'Arbresle, France) were used for brain slice electrophysiology. Horizontal brain slices containing the somatosensory cortex and the corresponding corticostriatal projection field were prepared according to the methods previously published (*Fino et al., 2005*). Horizontal brain slices with a thickness of 330 μm were prepared from rats (P$_{(20-30)}$) using a vibrating blade microtome (VT1200S, Leica Micosystems, Nussloch, Germany). Brains were sliced in a 95% CO2/5% O2-bubbled, ice-cold cutting solution containing (in mM) 125 NaCl, 2.5 KCl, 25 glucose, 25 NaHCO$_3$, 1.25 NaH$_2$PO$_4$, 2 CaCl$_2$, 1 MgCl$_2$, 1 pyruvic acid, and then transferred into the same solution at 34°C for one hour and then moved to room temperature.

Patch-clamp recordings

Patch-clamp recordings were performed as previously described (*Fino et al., 2010*; *Paillé et al., 2013*; *Cui et al., 2015*). For whole-cell recordings borosilicate glass pipettes of 4-6 MΩ resistance contained (in mM): 105 K-gluconate, 30 KCl, 10 HEPES, 10 phosphocreatine, 4 ATP-Mg, 0.3 GTP-Na, 0.3 EGTA (adjusted to pH 7.35 with KOH). The composition of the extracellular solution was (mM): 125 NaCl, 2.5 KCl, 25 glucose, 25 NaHCO$_3$, 1.25 NaH$_2$PO$_4$, 2 CaCl$_2$, 1 MgCl$_2$, 10 μM pyruvic acid bubbled with 95% O$_2$ and 5% CO$_2$. Signals were amplified using EPC10-2 amplifiers (HEKA Elektronik, Lambrecht, Germany). All recordings were performed at 34°C using a temperature control system (Bath-controller V, Luigs&Neumann, Ratingen, Germany) and slices were continuously superfused at 2–3 ml/min with the extracellular solution. Slices were visualized on an Olympus BX51WI microscope (Olympus, Rungis, France) using a 4x/0.13 objective for the placement of the stimulating electrode and a 40x/0.80 water-immersion objective for localizing cells for whole-cell recordings. Series resistance was not compensated. Current-clamp recordings were filtered at 2.5 kHz and sampled at 5 kHz and voltage-clamp recordings were filtered at 5 kHz and sampled at 10 kHz using the Patchmaster v2x32 program (HEKA Elektronik).

Chemicals

Chemicals were bath-applied or injected only in the recorded postsynaptic neuron through the patch-clamp pipette. DL-s2-amino-5-phosphono-pentanoic acid (D-AP5, 50 μM) (Tocris, Ellisville, MO, USA) was dissolved directly in the extracellular solution and bath applied. N-(piperidin-1-yl)-5-(4-iodophenyl)-1-(2,4-dichlorophenyl)-4-methyl-1H-pyrazole-3-carboxamide (AM251, 3 μM) (Tocris) and cyclosporin A (1 μM) (Tocris) were dissolved in ethanol and then added in the external solution at a final concentration of ethanol of 0.01–0.1%. 4-[Bis(1,3-benzodioxol-5-yl)hydroxymethyl]-1-piperidinecarboxylic acid 4-nitrophenyl hydrate (JZL184 hydrate, 1.5 μM) (Sigma), 2-arachidonoylglycerol (2-AG, 100 μM or 20 μM for puff experiments) (Tocris) and KT5720 (1 μM) (Tocris) were dissolved in DMSO and then added in the external solution at a final concentration of DMSO of 0.0025–0.1%. 4-[(2S)-2-[(5-isoquinolinylsulfonyl) methylamino]-3-oxo-3-(4-phenyl-1-piperazinyl)propyl] phenyl isoquinolinesulfonic acid ester (KN-62, 3 μM) (Tocris) was dissolved in DMSO and then added in the external solution at a final concentration of DMSO of 0.003%. KT5720 (1 μM) (Tocris) were dissolved in DMSO and applied internally via the patch-clamp pipette at a final concentration of DMSO of 0.1%. iPKI 6–22 (20 μM) (Tocris) was dissolved in 20% acetonitrile and applied intracellularly via the patch-clamp pipette.

Local applications of 2-AG were performed through a patch-clamp pipette (4-5 MΩ) placed at the vicinity (~50 μm) of the recorded medium-sized spiny neurons (MSN) and linked to a Picospritzer II system (Parker, USA), which supplies repeatable pressure pulses.

Spike-timing-dependent plasticity induction protocols

Electrical stimulations were performed with a bipolar electrode (Phymep, Paris, France) placed in the layer 5 of the somatosensory cortex (*Fino et al., 2005*; *Fino et al., 2010*; *Cui et al., 2015*). Electrical stimulations were monophasic at constant current (ISO-Flex stimulator, AMPI, Jerusalem, Israel).

Currents were adjusted to evoke 50-200pA EPSCs. Repetitive control stimuli were applied at 0.1 Hz. STDP protocols consisted in pairings of pre- and postsynaptic stimulations with the two events separated by a specific temporal interval (Δt). The paired stimulations were applied at 1 Hz throughout the study except in *Figure 5* in which 0.1, 1.0, 2.5 and 4.0 Hz were tested. Presynaptic stimulations corresponded to cortical stimulations and the postsynaptic stimulation of an action potential evoked by a depolarizing current step (30 ms duration) in MSNs. MSNs were maintained all along the STDP experiments at a constant holding membrane potential which corresponds to their initial resting membrane potential (-75±0.5 mV, n=110). Thus, EPSCs during baseline or after STDP protocol were measured at the same membrane potential (in voltage-clamp mode); STDP pairings (performed in current-clamp mode) were conducted also at this same holding membrane potential. Neurons were recorded for 10 min during baseline and for at least 60 min after STDP protocol; long-term synaptic efficacy changes were measured from 45 to -50 min. Thirty successive EPSCs (at 0.1 Hz) were individually measured and then averaged. Variation of series resistance, measured every 10 sec all along the experiment, beyond 20% led to the rejection of the experiment. For pharmacology experiments, after recording of 10 min control baseline, drugs were applied in the bath. A new baseline with drugs was recorded after a time lapse of 10 min (to allow the drug to be fully perfused) for 10 min before the STDP protocol. Drugs were present until the end of the recording (except when specified for AM251*). In a subset of experiments (for i-KT5720 and i-PKI 6-22) drugs were applied intracellularly through the patch-clamp pipette. Once the cell was patched, drugs were allowed to diffuse into the cell during at least 15 min before starting recording of the baseline.

## Electrophysiological data analysis

Off-line analysis was performed using Fitmaster (Heka Elektronik) and Igor-Pro 6.37 (Wavemetrics, Lake Oswego, OR, USA). Statistical analysis was performed using Prism 5.0 software (San Diego, CA, USA). In all cases 'n' refers to the number of repetitions of an experiment from single slice (each experiment being performed on different brain slices). All results were expressed as mean±s.e.m in the text and, for visualization purposes, as mean±s.d in the figures, and statistical significance was assessed using two-sided Student's t test or the one sample t test when appropriate at the significance level (p) indicated. We used the D'Agostino & Pearson omnibus normality test to test if the values come from a Gaussian distribution. All experimental data passed the normality test, except when indicated in the figure captions (where normality was assumed).

## Mathematical model
## Equations of the Mathematical Model

The dimensions and values of all the parameters are given in *Supplementary file 1A-C*. A full implementation can directly be downloaded from the ModelDB database (http://senselab.med.yale.edu/modeldb/), accession #187605. The model accounts for the signaling network depicted in *Figure 2A*, that gathers previous pharmacological evidence on STDP in MSN (*Shen et al., 2008*; *Pawlak and Kerr, 2008*; *Fino et al., 2010*).

### Synaptic plasticity and synaptic weights

In the model, we considered that the relative change in EPSC amplitude (synaptic weight, $W_{total}$) is the product of a pre- ($W_{pre}$) and a postsynaptic ($W_{post}$) component: $W_{total}=W_{pre}W_{post}$. To implement our hypothesis of 2-AG dependent presynaptic plasticity, in the lack of detailed information on the presynaptic signaling pathways relating eCB signaling to plasticity, we choose a simple phenomenological mechanism. Essentially, we adapted the mechanism developed to describe the control of plasticity by calcium concentrations in *Shouval et al. (2002)*, assuming instead that it is the amount of activation CB1R that controls $W_{pre}$:

$$\Omega(y_{CB1R}) = \begin{cases} 1 & \text{if} \quad y_{CB1R} < \theta_{LTD}^{start} \text{or} y_{CB1R} \in ]\theta_{LTD}^{stop}, \theta_{LTP}^{start}[ \\ 1 - A_{LTD} & \text{if} \quad y_{CB1R} \in [\theta_{LTD}^{start}, \theta_{LTD}^{stop}] \\ 1 + A_{LTP} & \text{if} \quad y_{CB1R} > \theta_{LTP}^{start} \end{cases} \quad (1)$$

where the function $\Omega$ sets the direction of plasticity (LTP, LTD or no plasticity); $y_{CB1R}= k_{CB1R} x_{CB1R}+ D_1$ describes the total eCB-dependent activation of the presynaptic signaling involved in plasticity and will be referred to as 'CB1R activation' below; $x_{CB1R}$ is the fraction of open CB1R (see below);

$D_1$ is a constant that accounts for presynaptic plasticity modulation by, for example, tonic dopamine; the $\theta$ s are the threshold levels of $y_{CB1R}$ determining plasticity induction; $A_{LTD}$ and $A_{LTP}$ are parameters determining the rate of LTD and LTP induction respectively. The dynamics of $W_{pre}$ is then given by the functions proposed in *Shouval et al. (2002)*:

$$\frac{dW_{pre}}{dt} = \frac{\Omega(y_{CB1R}) - W_{pre}}{\tau_{W_{pre}}(k_{CB1R}x_{CB1R} + D_2)}$$

$$\tau_{W_{pre}}(x) = \frac{P_1}{P_2^{P_3} + x^{P_3}} + P_4$$

(2)

$\tau_{W_{pre}}$ describes the time scale of presynaptic plasticity changes; $D_2$ is a constant that accounts for the modulation of plasticity time scales; $P_1$–$P_4$ are constants chosen to yield rapid changes of $W_{pre}$ for large *2-AG* values and very slow changes at very low *2-AG* (memory). To account for experimental observation that the presynaptic weight ranges from about 50 to 300%, $W_{pre}$ was clipped to 3.0.

The function $\Omega$ above (*Equation 1*) describes a sharp thresholding mechanism that we opted for for its simplicity in the absence of further supporting information. Smooth thresholding mechanisms can be used instead with no major alteration of our main results (see *Results*).

For $W_{post}$, we referred to the NMDAR signaling pathway. The molecular steps along this pathway are well characterized from Glutamate to CaMKII activation but the downstream molecular mechanisms, leading from CaMKII activation to changes of the synaptic weights are still unclear, especially in MSNs. Therefore, we adopted the hypothesis, already used in *Graupner and Brunel (2007)* and others before, that the long-term (steady state) increase of $W_{post}$ is proportional to the fraction of activated (phosphorylated) CaMKII. We assumed that $W_{post}$ increases linearly with the concentration of phosphorylated CaMKII subunits ($CaMKII_{act}$). Since the largest postsynaptic LTP we observed experimentally was about 450%, we set:

$$W_{post} = 1 + 3.5\frac{CaMKII_{act}}{CaMKII_{act}^{max}}$$

(3)

## CB1R activation and desensitization

We model CB1Rs activation with a simple three-state kinetic model: open ($x_{CB1R}$), desensitized ($d_{CB1R}$) and inactivated ($i_{CB1R}$):

$$\frac{dx_{CBIR}}{dt} = \alpha_{CBIR} \cdot eCB \cdot i_{CBIR} - (\beta_{CBIR} + \gamma_{CBIR})x_{CBIR}$$

$$\frac{dd_{CBIR}}{dt} = -\varepsilon_{CBIR}d_{CBIR} + \gamma_{CBIR}x_{CBIR}$$

$$x_{CBIR} + d_{CBIR} + i_{CBIR} = 1$$

(4)

where *eCB = 2-AG + 0.10 AEA* accounts for the fact that AEA is a partial agonist of CB1R (*Piomelli, 2003*). We assumed here that AEA is 10-times less efficient than *2-AG*; $\alpha_{CB1R}$, $\beta_{CB1R}$, $\gamma_{CB1R}$ and $\varepsilon_{CB1R}$ are the rate constants for the transitions between states.

## Postsynaptic element

We modeled the postsynaptic element as an isopotential compartment with membrane potential $V$ that varies according to:

$$C_m\frac{dV}{dt} = -g_L(V - V_L) - I_{AMPA}(V) - I_{NMDA}(V, G(t)) - I_{VSCC}(V) - I_{TRPV1}(V, AEA) - I_{act}(t)$$

(5)

$g_L$ and $V_L$ are leak conductance and reversal potential respectively; $I_{AMPA}$, $I_{NMDA}$, $I_{VSCC}$ and $I_{TRPV1}$ are currents through AMPAR, NMDAR, VSCC and TRPV1R, respectively; $I_{act}$ is the action current accompanying the postsynaptic (somatic) stimulation (back-propagating action potential on top of a step-like depolarization) and is described below; $G$ is the glutamate concentration in the synaptic cleft and *AEA* denotes anandamide concentration. NMDAR and AMPAR were modeled with two-state kinetic models and 1.0 mM $Mg^{2+}$ (*Destexhe et al., 1995*). L-type VSCCs are the main type of activated VSCCs in MSNs (*Carter and Sabatini, 2004*). We thus modeled VSCC currents using the

model and parameters of the Ca$_v$1.3 currents (*Wolf et al., 2005*). We added TRPV1 current because blocking it inhibits eCB-dependent LTP (*Cui et al., 2015*). The TRPV1 current, including its dependence on AEA, was modeled as:

$$I_{TRPV1}(V, AEA) = g_{TRPV1} \cdot V \cdot P_{TRPV1}^{open}(V, AEA) \tag{6}$$

where $g_{TRPV1}$ is maximal conductance of TRPV1. The mathematical expression for the probability of TRPV1 to be in the open state, $P_{TRPV1}^{open}$ was taken from *Matta and Ahern (2007)*. Note that from a modeling perspective, the TRPV1 current can be ignored. The resulting model would essentially yield the same output as those presented below as long as absence of TRPV1 is compensated for by a slight increase of NMDAR or VSCC conductances.

To model the dynamics of the cytoplasmic concentration of Calcium, *C*, we transform the currents with a calcium component in *Equation 5* to calcium fluxes by multiplying each of them by corresponding coefficient $\xi_x$; where *x* is *NMDA*, *VSCC*, or *TRPV1*. We moreover take into account the dynamics resulting from calcium exchange with internal calcium stores (Calcium-Induced Calcium Release, CICR). The equation for cytosolic calcium reads:

$$T_C(C)\frac{dC}{dt} = J_{IP_3R} - J_{SERCA} + J_{leak} - \frac{C - Ca_b}{\tau_{Ca_b}} - \xi_{NMDA}I_{NMDA} - \xi_{VSCC}I_{VSCC} - \xi_{TRPV1}I_{TRPV1} \tag{7}$$

where $J_{IP3R}$, $J_{SERCA}$, $J_{leak}$ are fluxes that describe CICR according to the model of *De Pittà et al. (2009)*. Note that in *De Pittà et al. (2009)*, the amount of active CaMKII (for IP$_3$ phosphorylation) is approximated by a simple Hill function of cytoplasmic calcium (their equation 11–12). Here, we used the more complex activation model from *Graupner and Brunel (2007)* (see below) to model CaMKII activation. In *Equation 7*, $Ca_b$ is the basal cytosolic calcium level resulting from equilibration with calcium diffusion out of the cell and $\tau_{Ca_b}$ the corresponding time scale. The presence of endogenous calcium buffer B (considered in quasi-equilibrium with cytosolic calcium at each time point) results in a calcium-dependent time scaling factor:

$$T_x(x) = 1 + \frac{B_T}{K_{dB}(1 + x/K_{dB})} \tag{8}$$

where $B_T$ and $K_{dB}$ are constants and x = *C* or $C_{ER}$. $C_{ER}$, the Calcium concentration in the endoplasmatic reticulum (ER) is given by

$$T_{C_{ER}}(C_{ER})\frac{dC_{ER}}{dt} = -\rho_{ER}(J_{IP_3R} - J_{SERCA} + J_{leak}) \tag{9}$$

where $\rho_{ER}$ is the ER to cytoplasm volume ratio.

CaMKII phosphorylation was modeled according to *Graupner and Brunel (2007)*. Originally, this model was developed to simulate hippocampal STDP, whereas our study is targeted to the striatum and MSNs in which protein phosphatase-1 (PP1) is inhibited by the striatum-specific subunit PPP1R1B (DARPP-32) rather than subunit PPP1R1A (Inhibitor-1). However, for the sake of simplicity (and since postsynaptic dopamine signaling is not explicitly considered here), we kept the equations and most of the parameters of *Graupner and Brunel (2007)*. Our major change concerns PKA activation by calcium. Beyond the expression of DARPP-32, another specificity of MSNs is to express the B72 regulatory subunit of PP2A at high levels in lieu of B56. This striatum-specific regulatory subunit provides PP2A with calcium-activation properties (*Ahn et al., 2007*). Therefore, calcium elevations in MSNs are expected to activate B72-PP2A. Active PP2A then can dephosphorylate DARPP-32 (at Tyr75), which would in turn disinhibit PKA. This process results in an effective activation of PKA by calcium, that is modeled in the *Graupner and Brunel (2007)* model via a Hill function of calcium with exponent $n_{PKA}$=8. More recent experimental evidence (*Ahn et al., 2007*) rather points to a lower value (2–3). We therefore changed for $n_{PKA}$=3.

2-AG production occurs in the postsynaptic neuron where it is initiated by DAG production via mGluR- and M1R-activated PLCβ. DAG-Lipase α (DAGLα) then produces 2-AG from DAG. DAG is co-produced together with IP3 by PLCβ (thus follows the same production dynamics as IP3) and is consumed by DAGLα (yielding 2-AG) and DAG kinase (DAGK, yielding phosphatidic acid):

$$\frac{dDAG}{dt} = R_P(C, IP_3, G) - \frac{r_{DGL} \cdot DAGL \cdot \varphi_{DAGL} \cdot DAG}{DAG + K_{DAGL}} - r_{DAGK} \cdot DAG \tag{10}$$

where $R_P(C, IP_3, G)$ is the term that describes IP$_3$ production dynamics in *De Pittà et al. (2009)*; IP$_3$ is IP$_3$ concentration; $\varphi_{DAGL}$ represents the fraction of activated DAGLα and *DAGL* its total (activated + not activated) concentration (see below), $r_{DGL}$ its maximal rate and $K_{DAGL}$ its Michaelis constant for DAG; $r_{DAGK}$ is the degradation rate by DAGK (that we assume linear for simplicity). 2-AG dynamics is obtained as the balance between postsynaptic synthesis (by DAGLα) and presynaptic degradation (by MAG-Lipase) upon retrograde transfer:

$$\frac{d2AG}{dt} = \frac{r_{DGL} \cdot DAGL \cdot \varphi_{DAGL} \cdot DAG}{DAG + K_{DAGL}} - r_{MAGL} 2AG \tag{11}$$

where $r_{MAGL}$ lumps together both enzyme degradation by MAG-Lipase and 2-AG spillover out of the synapse.

How DAGLα is activated in vivo is unknown, except for the calcium-dependence of its activation. We assumed DAGLα activation to rely on a single calcium activation step, modeled as a sigmoid function of the calcium concentration ($DAGL + n_c \text{Ca} \longleftrightarrow DAGL*$). Hence, the dynamics of the activated fraction of DAGL, $\varphi_{DAGL}$ is modeled here by:

$$\frac{d\varphi_{DAGL}}{dt} = r_k C^{n_c}(1 - \varphi_{DAGL}) - r_p \varphi_{DAGL} \tag{12}$$

where $n_c$, $r_k$ and $r_p$ are the constants of DAGL calcium-activation.

In vitro experiments suggest that DAGLα activation could be triggered by phosphorylation by a kinase (*Rosenberger et al., 2007*). An alternative, more complex, activation scheme could be that DAGLα is activated by a calcium-dependent kinase. Implementing this mechanism in our model (together with DAGL deactivation by a phosphatase) does not appreciably alter the results presented below.

Finally, to model AEA synthesis, we considered the well-documented 2-step pathway (*Starowicz et al., 2007*): $PE + PC \longrightarrow^{AT} NAPE \longrightarrow^{PLD} AEA$ with PE: phosphatidylethanolamine, PC: phosphatidylcholine, AT: N-acetyltransferase, NAPE: N-Arachidonyl-Phosphatidyl-ethanolamine and PLD: NAPE-selective phospholipase D. Note that alternative synthesis pathways exist, but because their relevance to neurons and MSNs is not clear yet (*Starowicz et al., 2007*), we did not consider them here. NAPE synthesis was modeled under the assumptions that *i*) PC and PE are in excess amounts and *ii*) the Ca$^{2+}$ concentrations necessary to reach half-maximal activation of AT (around 0.2 to 0.5 mM, *Hansen et al., 1998*) are well above the largest calcium levels in the model. Under those assumptions, $dNAPE/dt = v_{AT}C - r_{PLD}NAPE/(K_{PLD} + NAPE)$ where $v_{AT} = r_{AT}[PE][PC]/K_{act}$ with $r_{AT}$ the maximal rate of AT and $K_{act}$ its calcium activation constant. Likewise, the second step (AEA production) was modeled as $dAEA/dt = r_{PLD}NAPE/(K_{PLD} + NAPE) - r_{FAAH}AEA/(K_{FAAH} + AEA)$ where the latter summand represents AEA degradation by FAAH. Now, AEA synthesis is expected to proceed at a much faster rate than NAPE synthesis, so that NAPE is found at very low levels in cells (*Hillard et al., 1997*). The corresponding quasi-steady state assumption on NAPE concentration ($dNAPE/dt \approx 0$) then simplifies the expression of AEA dynamics to a single equation:

$$\frac{dAEA}{dt} = v_{AT}C - r_{FAAH}\frac{AEA}{K_{FAAH} + AEA} \tag{13}$$

where $r_{FAAH}$ and $K_{FAAH}$ represent FAAH enzyme activity and its Michaelis-Menten constant, respectively.

## Stimuli

After each presynaptic spike at time $t_{pre}^i$, we model the time course of glutamate (*G*) as a single exponential decay with peak value $G_{max}$ and clearance rate $\tau_G$:

$$G(t) = G_{max}\sum_i \exp\left(-\frac{t - t_{pre}^i}{\tau_G}\right)H(t - t_{pre}^i) \tag{14}$$

where $H(x)$ is the Heaviside function $H(x)=1$ if $x \geq 0$, 0 else. To model postsynaptic action current back-propagating from the soma, we use the sum of DC component of the current arising from the step-depolarization and a spike-induced transient that decays exponentially:

$$I_{act}(t) = -DC_{max} \sum_i \Pi(t, t^i_{post}, DC_{dur}) - AP_{max} \sum_i H(t - \delta - t^i_{post}) exp\left(\frac{t - \delta - t^i_{post}}{\tau_{bAP}}\right) \tag{15}$$

$$\Pi(t, t_0, L) = H(t - t_0) - H(t - t_0 - L)$$

where $DC_{max}$ and $DC_{dur}$ are the amplitude and the duration of step-current; $AP_{max}$ is the amplitude of the action current producing bAP, $\delta$ is the delay between the outset of the step depolarization and that of the bAP and $\tau_{bAP}$ the time scale for bAP decay. The time difference between the onset of EPSC and peak depolarization of bAP is given by $\Delta t_{STDP} = t^i_{post} + \delta - t^i_{pre}$.

## Parameters

The values of a large part of the parameters implicated in intracellular dynamics, eCB dynamics or electrophysiology in the model are restricted by previous experimental measurements (see *Supplementary file 1A-C*). To estimate the values of the parameters for which we lack previous experimental constraints, we used the experimental data shown in *Figure 1* and in (*Cui et al, 2015*), that is, we optimized those parameter values so that the model emulates the synaptic weight changes triggered by STDP protocols with various spike timings $\Delta t_{STDP}$ and numbers of paired stimulations $N$.

## Numerics

The ordinary differential equations of the model were integrated numerically with the LSODA solver from the ODEPACK fortran77 library (compiled for python with f2py) with absolute and relative tolerances both equal to $10^{-7}$. Initial conditions were set to the steady-state of each variable in the absence of stimulation. Numerical integration proceeded until the synaptic weights reach stable values (typically observed around $t \approx 5$min after the end of the stimulation protocol), and we kept the final value of the pre- and postsynaptic weights to compute the total synaptic weight change due to the stimulation protocol. Note that we also take into account that the experimental precision on the spike-timing delay ($\Delta t_{STDP}$) is around 2 to 5 ms. To emulate this, the simulation results were averaged (blurred) over this time window using convolution of $W_{pre}$ and $W_{post}$ with a normalized Gaussian function with s.d. = 3 ms.

## Sensitivity analysis

We quantified the model sensitivity to variations of the 50 parameters $p_k \, k = 1 \ldots 50$ whose values are not experimentally constrained ('free parameters', listed in *Figure 4—figure supplement 1C*). To this end, we generated 2500 random parameter vectors $\mathbf{p}^j = \{p^j_k\}$ $j = 1 \ldots 2500$ by randomly sampling each component $p_k$ independently from an uniform distribution ranging from 0.1 to 1.9 of its best-fit value $p_{ref,k}$ (given in *Supplementary file 1A-C*). We partitioned the ($\Delta t_{STDP}$, $N_{pairings}$)-plane of *Figure 4A1* (below) as a grid of $N$ points and measured the mean squared distance $D$ for each vector $\mathbf{p}^j$ as:

$$D(\mathbf{p}^j) = \left(1/N \sum_{i=1}^{N} \left(W_{pre}(i|\mathbf{p}^j) - W_{pre}(i|\mathbf{p}_{ref})\right)^2 + \left(W_{post}(i|\mathbf{p}^j) - W_{post}(i|\mathbf{p}_{ref})\right)^2\right)^{1/2} \tag{16}$$

where $W_{pre}(i|\mathbf{p}^j)$ denotes the value of the presynaptic weight at point $i$ of the ($\Delta t_{STDP}$, $N_{pairings}$)-grid when the values of the free parameters are given by the vector $\mathbf{p}^j$, and $\mathbf{p}_{ref}$ denotes the best-fit values. We then fitted the resulting points with linear regression

$$D(\mathbf{p}) = \mathbf{p}^T \mathbf{b} + b_0 \tag{17}$$

using ordinary least squares. Here $\mathbf{b} = (b_1, \ldots, b_M)$ is the vector of regression coefficients and $b_0$ a constant. Note that we did not adapt the parameters of numerical integration of the model to each set of randomly chosen parameters. Parameters that led to integration failure were thus not taken

into account. These rejections did not compromise uniformity of the distribution. We then computed for each parameter $p_k$ its standardized linear-regression coefficient (SRC)

$$SRC_k = b_k \frac{Var(p_k)}{Var(D)} \tag{18}$$

where $Var(p_k) = 0.27\, p_{\text{ref},k}{}^2$. $SRC_k$ is a measure of sensitivity of parameter $k$ (*Saltelli et al., 2002*): when $p_k$ varies away from its best-fit value, the distance between the resulting model output and the reference output of figure 4A1 is proportional to $SRC_k$.

## Additional information

### Funding

| Funder | Grant reference number | Author |
| --- | --- | --- |
| Agence Nationale de la Recherche | DopaciumCity | Laurent Venance Hugues Berry |

The funders had no role in study design, data collection and interpretation, or the decision to submit the work for publication.

### Author contributions
YC, IP, Acquisition of data, Analysis and interpretation of data, Drafting or revising the article; HX, Acquisition of data, Analysis and interpretation of data; BD, Analysis and interpretation of data, Contributed unpublished essential data or reagents; SG, Analysis and interpretation of data, Drafting or revising the article, Contributed unpublished essential data or reagents; LV, HB, Conception and design, Acquisition of data, Analysis and interpretation of data, Drafting or revising the article

### Author ORCIDs
Hao Xu, http://orcid.org/0000-0003-0695-8423

### Ethics
Animal experimentation: All experiments were performed in accordance with local animal welfare committee (Center for Interdisciplinary Research in Biology and EU guidelines, directive 2010/63/EU).

## Additional files

### Supplementary files
• Supplementary File 1. Parameters of the mathematical model
• Supplementary File 2. Modeling smooth thresholds for eCB-dependent plasticity

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
