## [Decision Letter]

Thank you for submitting your work entitled "Spike-timing-dependent dynamics of
2-arachidonoylglycerol gates endocannabinoid-mediated LTP and LTD" for
consideration by *eLife*. Your article has been reviewed by three peer
reviewers, one of whom is a member of our Board of Reviewing Editors. The evaluation has
been overseen by the Reviewing Editor and Eve Marder as the Senior Editor.

The reviewers have discussed the reviews with one another and the Reviewing Editor has
drafted this decision to help you prepare a revised submission.

Summary:

This paper proposes a model involving pre and postsynaptic signaling components involved
in corticostriatal synaptic plasticity. Key components of the model include presynatpic
endocannabinoid signaling with three thresholds for different forms of plasticity, and a
signaling network depending on calcium for the postsynaptic network. The paper is
distinctive in presenting several high-level tests of the signaling model, including
some non-obvious predictions. The authors present MAGL activity as a key controller of
synaptic plasticity, and suggest that endocannabinoid receptors may be important in
bidirectional regulation of plasticity.

Essential revisions:

1) The reviewers appreciated the experimental and theory combination of approaches to
study plasticity.

2) The model assumes a number of model free parameters such as sharp thresholds for
plasticity. The authors should test model outcomes of shallow thresholds to see if the
results still hold, and also conduct parameter sensitivity analysis to examine how
sensitive these parameters are.

3) The predictions should extend to a somewhat broader range of stimulus cases, such as
modification of pairing frequencies and number.

4) All reviewers had questions on the details of the model pathways and role of
sub-parts of the overall model. The authors should clarify this, with special attention
to stating which pathways are really needed for which outcome of the model.

*Reviewer #1*:

1) A key assumption in the model is the presence of three thresholds for LTD start, LTD
stop, and LTP start, for the endocannabinoid system. The key step here (thresholding) is
not implemented biophysically or biochemically, but by a mathematical threshold. I think
this weakens their case for a mechanistic account of STDP. The authors discuss this and
have a schematic in Figure 9—figure supplement 1, but I don't feel that the proposed
sharp thresholds are physiological. I would have liked to see a chemical implementation
of the thresholds. Specifically, I am concerned that the presence of a shallow (more
chemical-like) turn on rather than a sharp all-or-none threshold may invalidate the
results. The authors should address this.

2) A related point from Figure 3: The
threshold positioning and the extra bump by CB1R at the start is quite finicky. I am
dubious about the dependence on such fine-tuning. Even a small shift of either the
threshold or the response would invalidate the prediction that such a mechanism could
account for the properties of eCB-tLTP. Here, in fact, a shallower activation function
to replace the threshold might prove to be more robust, but less precise.

3) Figure 4 is impressive in its match to
experiments. I wonder if it is possible for the authors to achieve a similar match for
Figure 6,Figure 7,Figure 8, which also test a
series of model predictions. All look good but the prediction is just a single
time-point. In all cases it would be nice if the simulations went the extra step to
match the experimental time-course, rather than just predict amplitude of LTP or
LTD.

4) The authors should comment on what stochasticity would do to their analysis. I am
concerned that this would further weaken any analysis depending on sharp thresholds.

5) The CaMKII activity seems to be a bit of an orphan in the study. It turns on for
sufficient pairings for pre-post pairings, but how does this impact the plasticity? I
was not able to clearly see where this happened.

6) The authors get CaMKII to turn on (Figure 2)
but I don't see that the LTD stimulus (or any other) gets it to turn off. This seems
incomplete.

7) ModelDB does not seem to have this model. It isn't really possible to assess the
implementation without it. The authors should present the model and any simulation files
needed to generate the figures, as supplementary material.

Reviewer #2:

Cui et al. use a phenomenological, but quantitative model to increase the understanding
of activity patterns leading to LTP/LTD. The model predictions agree with experiments.
The main contribution is to illustrate how 2AG can control both LTP and LTD.

There are things which can strengthen the study, and make the model more transparent.
The latter issue is important as others then can further improve it.

1) The experimental paradigm builds on repetition with 1 Hz pre- and postsynaptic
pairings in different orders/delays. As the model mechanisms for 'many-pairing' LTP
builds on successive activation of CamKII while the 'many-pairing' LTD builds on 2AG
production with presynaptic effects. For the latter, postsynaptic depletion of ER_Ca and
presynaptic receptor desensitization contribute, thus it would be interesting to see
what is predicted if 1 Hz pairing frequency is modified to e.g. 2 or 0.1 Hz. From Figure 3 the ER depletion seems significant after
only 20 pairings, is that realistic? Also is this depletion needed for the model to
work?

2) It seems that the model free parameters are fitted to reproduce experimental outcome
(see subheading “Parameters”), thus the model is not predicting the LTP and LTD results
(since the LTP and LTD outcome is used to tune the model), but rather the model can work
as a quantitative hypothesis on important subcellular mechanisms. Please specify which
model parameters are considered free and their sensitivity to variations.

3) Details on the model that need to be made more transparent are, for example:

A) The activation of DAGL. A Ca dependent phosphorylation reaction assumed but is it not
a more direct Ca activation of DAGL and presence of DAG sufficient to produce 2AG for
example? Likely no consequences for the outcome but please clarify;

B) Which and how different Ca sources (NMDA_Ca, Ca via TRPV1R, ER_Ca, L_Ca, etc.)
contribute to the total pool of Ca used to activate CamKII, DAGL, etc. (add e.g. a
supplementary figure following Figure 2 or Figure 3 showing how the total Ca elevation is the
sum of several sources). How is the 'unintuitive' result achieved that Ca is larger if
post-pre stimulation is used compared to pre-post? If only NMDA_Ca is considered
pre-post should give rise to more Ca influx into the cell as compared to post-pre;

C) Please plot separately the *W*_pre_ and
*W*_pro_ to see how it compares to
*W*_total_ plotted in several figures;

D) In paragraph three, subheading “Postsynaptic element” the postsynaptic model from
Graupner and Brunel is adjusted for MSNs and it is assumed that PKA is indirectly
activated by Ca via PP2A, motivate this or maybe remove the Ca dependency of PKA as it
is hard to see how the AC5-PKA reactions is effectively stimulated by Ca in MSNs, even
in the presence of DARPP75 disinhibition via PP2A). That CaMKII is activated following a
sufficient number of pairings in MSNs is reasonable, but it probably happens in a
slightly different way as in hippocampus.

---

## [Author Response]

*Essential revisions:*

*1) The reviewers appreciated the experimental and theory combination of
approaches to study plasticity.2) The model assumes a number of model free parameters
such as sharp thresholds for plasticity. The authors should test model outcomes of
shallow thresholds to see if the results still hold, and also conduct parameter
sensitivity analysis to examine how sensitive these parameters are.*

The revised manuscript includes two new sets of results to account for this point.

1) We implemented an alternative version of the model, where the sharp plasticity
thresholds are replaced by a smooth function based on combinations of Hill functions.
For clarity of the main text, the exact equation of this function is given in the new
[Supplementary-material SD2-data].
In this function, the sharpness of the plasticity transition is controlled by a
parameter *k_S_*. The new Figure 4—figure supplement 1 (panel B) shows that model output is essentially
conserved with this smooth threshold function, *even though we did not change any
of the model parameters outside the threshold function*. We illustrate this
for *k_S_* =2 on Figure 4
but other values of *k_S_* essentially lead to the same
conclusion. Therefore our choice of a sharp thresholding for eCB-dependent plasticity is
not crucial for the model output.

2) We now provide in new Figure 4—figure supplement 1 (panel C) a proper analysis for the sensitivity of the parameters, using
standardized linear-regression coefficients (see Methods). As expected, the most
sensitive parameters are those related to reactions that are known from pharmacological
experiments to be indeed crucial to STDP and the dynamics of CB1R desensitization, in
agreement with the importance of CB1R desensitization in the decay of eCB-LTP above
15-20 post-pre stimulations. More surprising is the sensitivity of the model to the
dynamics of glutamate in the synaptic cleft. We therefore conclude that alterations of
the dynamics of glutamate release and uptake could play an important role in the control
of STDP at the corticostriatal synapse.

Changes made: Figure 4—figure supplement 1
shows the results with smooth threshold (panel B) and sensitivity analysis (panel C).
Both results (threshold smoothness and sensitivity) are presented is a new subsection of
the revised version (paragraphs 2-4, subheading “The mathematical model accounts for
bidirectional eCB- and NMDAR- mediated STDP”). The definition of the smooth threshold
function is given in the new [Supplementary-material SD2-data] A new subsection of the Methods section explains the
methodology used for sensitivity analysis ("Sensitivity analysis"). We also
added new text on the choice of sharp thresholds in Methods (subheading “Synaptic
plasticity and synaptic weights”).

*3) The predictions should extend to a somewhat broader range of stimulus cases,
such as modification of pairing frequencies and number.*

Here too, we have taken this point into account with two new sets of results:

1) The revised version features a new set of modeling and experimental results obtained
by varying both the spike timing and the pairing frequency (from 0.1 Hz to 4 Hz) with 10
pairings. Those results (experimental and modeling) are shown in the new Figure 5. The main prediction of the model are that,
when frequency increases above 1Hz, the tLTP triggered by post-pre stimulations (ΔtSTDP
< 0) persists and is even observed for an increasingly large ΔtSTDP range.
Unexpectedly, the model also predicts the expression of another tLTP, triggered by 10
pre-post stimulations (ΔtSTDP > 0) for frequency larger than 2Hz. We present an
entire new set of experiments at 0.1, 1.0, 2.5 and 4.0 Hz that validates these
predictions (Figure 5). In terms of
pharmacology, we also show in the supplementary figure of Figure 5 that again, model and experiments agree about the fact
that the tLTP at high frequencies (4 Hz) becomes of mixed origin, both eCB- and
NMDAR-dependent.

2) Regarding the modification of pairing numbers, we now show in the panel A of the new
Figure supplement of Figure 4, what is the output
of the model when more than 100 pairings are used. The behavior observed with 100
pairings is conserved for larger pairings numbers.

Changes made: Figure 4—figure supplement 1
shows the model output with > 100 pairings (panel A, presented in the Results
section, subheading “The mathematical model accounts for bidirectional eCB- and NMDAR-
mediated STDP”). The frequency-dependence results (model and experiments) are shown in a
new figure (Figure 5) and commented in a new
subsection of the Results section ("Frequency dependence of eCB-tLTP") and
discussed in the new Discussion section (paragraphs five and six).

*4) All reviewers had questions on the details of the model pathways and role of
sub-parts of the overall model. The authors should clarify this, with special
attention to stating which pathways are really needed for which outcome of the
model.*

We give a detailed reply to those points in our response to the reviewers below. But to
summarize:

1) We simplified the model at the level of DAG-Lipase activation. In the original
version, DAG-Lipase activation by calcium occurred via DAG-Lipase phosphorylation
catalyzed by a kinase (and the reverse phosphatase), that was itself calcium-activated.
In the revised version, we replaced this unnecessary complex scheme by direct activation
of DAG-Lipase by calcium (equation 12,). Note that all the model figures in the article
have been revised to correspond to this new, simplified model.

2) We kept TRPV1 in the model because we have shown in a previous paper (Cui et al.,
2015) that blocking TRPV1 inhibits eCB-dependent LTP. However, from a modeling
perspective, the TRPV1 current can be ignored. The resulting model would essentially
yield the same output as obtained with the standard model, as long as absence of TRPV1
is compensated for by a slight increase of NMDAR or VSCC conductances. We have added new
text regarding this point in the Methods section (subheading “Postsynaptic
element”).

3) The importance of each separate pathway can now be better judged through the
sensitivity analysis of the model parameters featured in the revised version (see our
reply to point 1. above). In particular, our interpretation of the sensitivity analysis
in the revised version clearly points to the most important elements of pathways
(paragraph four, subheading “The mathematical model accounts for bidirectional eCB- and
NMDAR- mediated STDP”).

Reviewer #1:

1) A key assumption in the model is the presence of three thresholds for LTD
start, LTD stop, and LTP start, for the endocannabinoid system. The key step here
(thresholding) is not implemented biophysically or biochemically, but by a
mathematical threshold. I think this weakens their case for a mechanistic account of
STDP. The authors discuss this and have a schematic in Figure 9—figure supplement 1,
but I don't feel that the proposed sharp thresholds are physiological. I would have
liked to see a chemical implementation of the thresholds. Specifically, I am
concerned that the presence of a shallow (more chemical-like) turn on rather than a
sharp all-or-none threshold may invalidate the results. The authors should address
this.

*2) A related point from Figure 3 C1: The
threshold positioning and the extra bump by CB1R at the start is quite finicky. I am
dubious about the dependence on such fine-tuning. Even a small shift of either the
threshold or the response would invalidate the prediction that such a mechanism could
account for the properties of eCB-tLTP. Here, in fact, a shallower activation
function to replace the threshold might prove to be more robust, but less
precise.*

In the revised version, we implemented an alternative version of the model, where the
sharp plasticity thresholds are replaced by a smooth function based on combinations of
Hill functions. For clarity of the main text, the exact equation of this function is
given in the new [Supplementary-material SD2-data]. In this function, the sharpness of the plasticity transition is
controlled by a parameter *k_S_*. New Figure 4—figure supplement 1 (panel B) shows that model output is
essentially conserved with this smooth threshold function, even though we did not change
any of the model parameters outside the threshold function. We illustrate this for
*k_S_* =2 on Figure 4 but other values of *k_S_* essentially lead to the
same conclusion. Therefore our choice of a sharp thresholding for eCB-dependent
plasticity is not crucial for the model output.

Changes made: Figure 4—figure supplement 1
shows the results with the smooth threshold (panel B). These results are presented is a
new subsection of the revised version (paragraph 3, subheading “The mathematical model
accounts for bidirectional eCB- and NMDAR- mediated STDP”). The definition of the smooth
threshold function is given in the new [Supplementary-material SD2-data]. We also added new text on the choice of
sharp thresholds in Methods (subheading “Synaptic plasticity and synaptic weights”).

3) Figure 4 is impressive in its match to
experiments. I wonder if it is possible for the authors to achieve a similar match
for Figure 6,Figure 7,Figure 8, which also test
a series of model predictions. All look good but the prediction is just a single
time-point. In all cases it would be nice if the simulations went the extra step to
match the experimental time-course, rather than just predict amplitude of LTP or
LTD.

We thank the reviewer for her/his nice comment on the match of Figure 4 with experiments. As pointed out by the reviewer, in its
current state, the model does a very good job in predicting the final (long-term)
amplitude of STDP. But it cannot reliably be used to predict the time course of the
synaptic weight after the pairings because our knowledge of many of the molecular
processes involved in plasticity maintenance and expression downstream of CaMKII and
CB1R are still not known with enough detail. For the NMDAR-CaMKII part, the signaling
pathway from glutamate to CaMKII activation is rather well characterized but the
molecular mechanisms leading from CaMKII activation to changes of the synaptic weights
are still unclear, especially in MSNs (direct AMPAR activation, alterations of AMPAR or
NMDAR trafficking, other modifications of the anchoring/scaffolding properties of the
PSD?). Likewise, for the mGluR-eCB-CB1R pathway it is not clear what are the molecular
targets that are set in motion following CB1R-triggered activation of PKA or/and
inhibition of VGCCs and that lead to changes of the presynaptic weight. In these
conditions, it would seem rather pointless to propose a model to predict the temporal
evolution of the synaptic weights. By contrast, our results strongly suggest the
validity of the assumption that the steady-state (long-term) values of the pre- and
post-synaptic weight is proportional to the amount of activated CB1R and CaMKII,
respectively, so that predicting the (long-term) amplitude of STDP is still
possible.

*4) The authors should comment on what stochasticity would do to their analysis.
I am concerned that this would further weaken any analysis depending on sharp
thresholds.*

We thank the reviewer for this very interesting comment. Actually, we have studied the
issue of stochasticity on STDP protocols. As a first approach, we have studied and
applied STDP protocols where spike timing, i.e. the time between pre and post pairings
is not deterministic/Dirac distributed but a random variable. With increasing variance
of these random spike timings and at 1 Hz, the model shows a striking differential
behavior: whereas NMDAR-LTP is very sensitive and disappears quickly when variance
increases, eCB-dependent plasticity (eCB-LTD and eCB-LTP) remains expressed for a much
larger amount of noise. This results is however also frequency-dependent since for
larger frequencies, NMDAR-LTP becomes much more resilient. Importantly, here again, the
predictions of our model have been validated by experiments, which confirm the major
trends of the model. Thus, we agree that stochasticity counter STDP expression in the
model, but more importantly: i) noise differentially affects eCB-dependent and
NMDAR-dependent plasticity and ii) this effect is also observed experimentally. We did
not include these results in the present version of this manuscript because we feel the
revised version already contains a large amount of information so adding the study of
stochasticity on top of it would only go against the clarity of the paper. In addition,
to be fully presented, these results (model + experiments) would require a full
paper.

5) The CaMKII activity seems to be a bit of an orphan in the study. It turns on
for sufficient pairings for pre-post pairings, but how does this impact the
plasticity? I was not able to clearly see where this happened.

6) The authors get CaMKII to turn on (Figure 2) but I don't see that the LTD stimulus (or any other) gets it to turn
off. This seems incomplete.

As explained above, the molecular mechanisms, leading from CaMKII activation to changes
of the synaptic weights are still unclear. In these conditions, we adopted the
hypothesis, already used by Graupner and Brunel, 2007 and others before, that the
long-term (steady state) increase of the post-synaptic weight is proportional to the
fraction CaMKII that is activated (phosphorylated). Considering the good match with our
experiments, this seems a posteriori a valid hypothesis. Regarding bidirectionality,
since our model for the NDMAR pathway is based on the model by Graupner and Brunel,
2007, it has inherited its potentiality to implement both LTP and LTD, depending on the
stimulation. However, we have been undertaking experimental studies of STDP of the
corticostriatal synapses in slices for some time now, but we never observed notable
NMDAR-dependent LTD (at least in "control" conditions, i.e. in the absence of
GABA or transporter blockers). This is the reason why we do not mention NMDAR-dependent
LTD in the manuscript.

Changes made: the Methods section now contains a clearer and longer statement about how
CaMKII was used to model post-synaptic plasticity (paragraph three, subheading
“Postsynaptic element”).

*7) ModelDB does not seem to have this model. It isn't really possible to assess
the implementation without it. The authors should present the model and any
simulation files needed to generate the figures, as supplementary material.*

The model has now been uploaded in the ModelDB database (accession # 187605).

Reviewer #2:

1) The experimental paradigm builds on repetition with 1Hz pre- and postsynaptic
pairings in different orders/delays. As the model mechanisms for 'many-pairing' LTP
builds on successive activation of CamKII while the 'many-pairing' LTD builds on 2AG
production with presynaptic effects. For the latter, postsynaptic depletion of ER_Ca
and presynaptic receptor desensitization contribute, thus it would be interesting to
see what is predicted if 1 Hz pairing frequency is modified to e.g. 2 or 0.1 Hz. From
Figure 3 the ER depletion seems
significant after only 20 pairings, is that realistic? Also is this depletion needed
for the model to work?

In order to take this comment into account, the revised version now features a new set
of modeling and experimental results obtained by varying both the spike timing and the
pairing frequency (from 0.1 Hz to 4 Hz) with 10 pairings. Those results (experimental
and modeling) are shown in the new Figure 5. The
main prediction of the model are that, when frequency increases above 1Hz, the tLTP
triggered by post-pre stimulations (ΔtSTDP < 0) persists and is even observed for
increasingly large ΔtSTDP ranges. Unexpectedly, the model also predicts the expression
of another tLTP, triggered by 10 pre-post stimulations (ΔtSTDP > 0) for frequency
larger than 2Hz. We carried out an entire new set of experiments at 0.1, 1.0, 2.5 and
4.0 Hz that validates (Figure 5) these
predictions. In terms of pharmacology, we also show in the supplementary figure of Figure 5 that again, model and experiments agree
about the fact that the tLTP at high frequencies (4 Hz) becomes of mixed origin, both
eCB- and NMDAR-dependent.

Changes made: This frequency-dependence results (model and experiments) are shown in a
new figure (Figure 5), commented in a new
subsection of the Results section ("Frequency dependence of eCB-tLTP") and
discussed in the new Discussion section (paragraph six).

*2) It seems that the model free parameters are fitted to reproduce experimental
outcome (see subheading “Parameters”), thus the model is not predicting the LTP and
LTD results (since the LTP and LTD outcome is used to tune the model), but rather the
model can work as a quantitative hypothesis on important subcellular mechanisms.
Please specify which model parameters are considered free and their sensitivity to
variations.*

The revised manuscript provides in the new Figure 4—figure supplement 1 (panel C), a proper analysis for the sensitivity of the
parameters, using standardized linear-regression coefficients (see Methods). As
expected, the most sensitive parameters are those related to reactions that are known
from pharmacological experiments to indeed be crucial to STDP and the dynamics of CB1R
desensitization, in agreement with the importance of CB1R desensitization in the decay
of eCB-LTP above 15-20 post-pre stimulations. More surprising is the sensitivity of the
model to the dynamics of glutamate in the synaptic cleft. We therefore conclude that
alterations of the dynamics of glutamate release and uptake could play an important role
in the control of STDP at the corticostriatal synapse.

Changes made: Figure 4—figure supplement 1
shows the results of our sensitivity analysis (panel C) and is presented in a new
subsection of the revised version (paragraph four, subheading “The mathematical model
accounts for bidirectional eCB- and NMDAR- mediated STDP”). A new subsection of the
Methods section explains the methodology used for sensitivity analysis
("Sensitivity analysis").

*3) Details on the model that need to be made more transparent are, for
example:*

A) The activation of DAGL. A Ca dependent phosphorylation reaction assumed but
is it not a more direct Ca activation of DAGL and presence of DAG sufficient to
produce 2AG for example? Likely no consequences for the outcome but please
clarify;

We thank the reviewer for her/his insightful comment. In the revised version, we have
simplified how DAG-Lipase is activated in the model. In the original version, DAG-Lipase
activation by calcium occurred via DAG-Lipase phosphorylation catalyzed by a kinase (and
the reverse phosphatase), that was itself calcium-activated. In the revised version, we
replaced this unnecessary complex scheme by direct activation of DAG-Lipase by calcium
(equation 12). Note that we could have pushed further in this simplification effort by
removing DAG-Lipase entirely and e.g. assuming a simple unimolecular reaction for 2-AG
production, such as DAG → 2-AG. However, since our previous results (Cui et al., 2015)
show that specific inhibition of DAG-lipase suppresses eCB-LTP, we decided to keep the
explicit account of DAG-Lipase activation in the revised model.

Changes made: All the model figures in the article have been replotted using this new,
simplified model.

B) Which and how different Ca sources (NMDA_Ca, Ca via TRPV1R, ER_Ca, L_Ca,
etc.) contribute to the total pool of Ca used to activate CamKII, DAGL, etc. (add
e.g. a supplementary figure following Figure 2
or Figure 3 showing how the total Ca elevation
is the sum of several sources). How is the 'unintuitive' result achieved that Ca is
larger if post-pre stimulation is used compared to pre-post? If only NMDA_Ca is
considered pre-post should give rise to more Ca influx into the cell as compared to
post-pre;

Author response image 1 illustrates the contribution of each calcium source to the total
intracellular calcium pool during a single post-pre (left) or pre-post (right)
pairing.

In both cases, the contribution of CICR is almost zero. Indeed, the effects of CICR
develop progressively as IP3 accumulates during the successive pairings and is not
sensible for the first stimulation. Contribution through TRPV1 is as well negligible
since TRPV1 needs AEA to open and AEA starts to be large enough for this only in the
next pairings. Calcium flux through VDCCs is important for post-pre pairings but also
for pre-post ones, since its absence annihilates the global calcium influx during the
bAP. Finally NMDAR is responsible for the majority of the calcium influx during the
EPSP. We propose not to include those results in the revised manuscript, because we feel
the revised version already contains a large amount of information and adding this
figure would too much interrupt the reading flow of the manuscript. But we would accept
to integrate it in the manuscript, in case the reviewer and editor find it
necessary.

C) Please plot separately the W_pre_ and W_pro_ to see how it
compares to W_total_ plotted in several figures;

We have modified Figure 4 (panels B and C) to
take this issue into account. Actually, in the control case, the information about
*W_pre_*and *W_post_*is contained in our in silico "Knock out models" of Figure 4. *W_post_*relies entirely on CaMKII activation, so our "NMDAR signaling
knockout" model (Figure 4B1) corresponds to a situation where the contribution of
*W_post_*is absent and only *W_pre_*contributes to
*W_total_*. Likewise, because
*W_pre_*depends on CB1R activation only, the "CB1R
knockout model" actually reflects the case where only *W_post_*contributes to *W_total_*.

Changes made: In Figure 4, the title of the
panels for the "NMDAR signaling KO" (Figure 4B1) and the "CB1R signaling
KO" models (Figure 4C1) now read "NMDAR signaling KO
(*W_pre_*only)" and "CB1R signaling KO
(*W_post_*only)", respectively. We have also added new text to explain this
(paragraph two, subheading “The mathematical model accounts for bidirectional eCB- and
NMDAR- mediated STDP”).

*D) In paragraph three, subheading “Postsynaptic element” the postsynaptic model
from Graupner and Brunel is adjusted for MSNs and it is assumed that PKA is
indirectly activated by Ca via PP2A, motivate this or maybe remove the Ca dependency
of PKA as it is hard to see how the AC5-PKA reactions is effectively stimulated by Ca
in MSNs, even in the presence of DARPP75 disinhibition via PP2A). That CaMKII is
activated following a sufficient number of pairings in MSNs is reasonable, but it
probably happens in a slightly different way as in hippocampus.*

For the NMDAR-CaMKII signaling pathway, the major difference between MSNs and
hippocampal neurons is indeed the enrichment of DARPP32 in MSNs in place of I-1 in
hippocampus. But another specificity of MSNs is that they express large amounts of the
B72 regulatory subunit of PP2A in lieu of the usual B56 one. This striatum-specific
regulatory subunit provides B72-PP2A with calcium-activation properties (see e.g. Ahn et
al., 2007). Therefore, calcium elevations in MSNs are expected to activate PP2A. Active
PP2A can then dephosphorylate DARPP-32 on its pThr75 site, which disinhibits PKA (see
e.g. the modeling studies by Lindskog et al., PLoS Comput Biol 2006 or Gutierrez-Arenas
et al., PLoS Comput Biol 2014).

Changes made: We have added new text to explain this phenomenon (subheading “Synaptic
plasticity and synaptic weight”).